# SPATIAL REASONING WITH VISION-LANGUAGE MODELS IN EGO-CENTRIC MULTI-VIEW SCENES

**Mohsen Gholami[1]\*, Ahmad Rezaei[1] , Zhou Weimin[2] , Sitong Mao[2] , Shunbo Zhou[2] ,
Yong Zhang[1]†, Mohammad Akbari[1]†**
[1] Huawei Technologies Canada, [2] Huawei Cloud

🌐 Project Page      Code      🤗 Ego3D-Bench

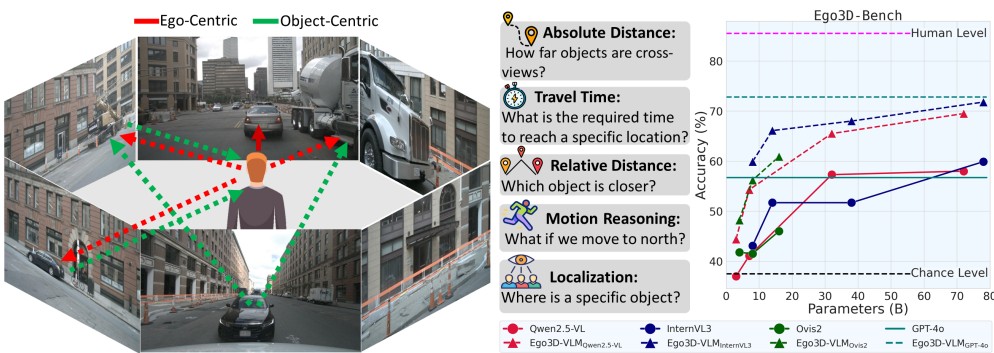

Figure 1: `Ego3D-Bench` is a 3D spatial benchmark for VLMs using ego-centric multi-view images. It spans ego- and object-centric perspectives across 5 categories. A significant gap exists between human and VLM performance; our method, *Ego3D-VLM*, consistently narrows this gap.

## ABSTRACT

Understanding 3D spatial relationships remains a major limitation of current Vision-Language Models (VLMs). Prior work has addressed this issue by creating spatial question-answering (QA) datasets based on single images or indoor videos. However, real-world embodied AI agents—such as robots and self-driving cars—typically rely on ego-centric, multi-view observations. To this end, we introduce `Ego3D-Bench`, a new benchmark designed to evaluate the spatial reasoning abilities of VLMs using ego-centric, multi-view outdoor data. `Ego3D-Bench` comprises over **8,600** QA pairs, created with significant involvement from human annotators to ensure quality and diversity. We benchmark 16 SOTA VLMs, including GPT-4o, Gemini1.5-Pro, InternVL3, and Qwen2.5-VL. Our results reveal a notable performance gap between human level scores and VLM performance, highlighting that current VLMs still fall short of human level spatial understanding. To bridge this gap, we propose *Ego3D-VLM*, a training-free framework that enhances 3D spatial reasoning of VLMs. *Ego3D-VLM* generates cognitive map based on estimated global 3D coordinates, resulting in **12%** and **56%** average improvements on multi-choice QA and absolute distance estimation, respectively. *Ego3D-VLM* can be integrated with any existing VLM. Together, `Ego3D-Bench` and *Ego3D-VLM* offer valuable tools for advancing toward human level spatial understanding in real-world, multi-view environments. Code is available in the supplementary materials.

---

\*Corresponding Author: mohsen.gholami1@huawei.com
†Co-Leads

# 1 INTRODUCTION

3D spatial understanding is a critical capability for embodied AI agents operating in the real world (Chaplot et al., 2021; Wang et al., 2024b). This includes perceiving the location of surrounding objects, estimating their distances, and reasoning about their motion. VLMs have recently emerged as powerful tools to integrate visual perception and language reasoning, making them promising components for building intelligent embodied AI systems (Ma et al., 2024b; Li et al., 2024b). Thus, enhancing and evaluating the spatial understanding of VLMs has become an increasingly important research direction (Team, 2025; Cheng et al., 2024; Chen et al., 2024; Zhao et al., 2025; Zhang et al., 2025b; Wang et al., 2024a; Wu et al., 2024b;a; Song et al., 2025; Ray et al., 2025).

Recent benchmarks on spatial understanding focus mainly on spatial reasoning from single images (Cheng et al., 2024; Chen et al., 2024; Liao et al., 2024) or videos captured in static indoor environments (Yang et al., 2024; Daxberger et al., 2025). In such cases, spatial understanding is framed around passive observations, where a single camera moves in a room to create a video, and the model should infer spatial relationships in a relatively static scene. This setup differs fundamentally from the perceptual experience of real-world embodied agents such as self-driving cars or mobile robots. These agents rely on ego-centric multi-view observations (Ma et al., 2024b), provided by multiple cameras simultaneously capturing front, side, and rear views of their surroundings. These views are not interchangeable or purely visual; they carry explicit spatial semantics tied to the agent's frame of reference. E.g, "left" & "right" refer to fixed directions relative to the agent's body, and must be interpreted consistently over time as the agent moves through dynamic environments. This distinction is crucial: while prior video-based datasets may offer multiple viewpoints, they do not reflect the structured, directional, and temporally evolving nature of ego-centric multi-view inputs. Also, existing benchmarks do not evaluate VLMs' reasoning ability across these spatially grounded perspectives in dynamic, real-world scenes.

This gap motivates the need for new benchmarks that better align with the spatial reasoning demands of embodied agents. To this end, we introduce `Ego3D-Bench`, a benchmark of 8.6K QA pairs carefully curated from the validation set of 3 public datasets: NuScenes (Caesar et al., 2020), Waymo Open Dataset (Sun et al., 2020), and Argoverse 1 (Chang et al., 2019). Human annotators played a central role in both the dataset construction and the rigorous quality review process to ensure the reliability of the benchmark. We focused specifically on ego-centric multi-view tasks, rather than building a general-purpose benchmark, to complement existing monocular spatial benchmarks (e.g., VSI-Bench (Yang et al., 2024)). Thus, we excluded questions that can be answered based on each view independently across multiple images (e.g., counting objects) or general knowledge of LLMs (e.g., size of well-known objects). `Ego3D-Bench` extends existing multi-view benchmarks (Yang et al., 2024; Yeh et al., 2025) to ego-centric scenario that is specialized for embodied AI.

We evaluated 16 SOTA VLMs, including generalist and 3D spatial ones on `Ego3D-Bench`, revealing a significant gap between human performance and current VLMs. We hypothesize that a key limitation lies in the inability of VLMs to construct a coherent world model from multi-view images. In contrast, humans naturally integrate visual info from their left, right, and front views into a unified spatial representation, enabling real-time reasoning and navigation. Prior work has attempted to bridge this gap by first generating a 3D point-cloud (Team, 2025; Huang et al., 2024; Hong et al., 2023b; Zhang et al., 2024a; Hong et al., 2023a) or rendering a bird-eye-view (BEV) image of the scene (Qi et al., 2024). Although point-clouds and BEV images offer rich spatial information, they are challenging to reconstruct in dynamic environments, struggle with sparse multi-view inputs, and significantly increase inference time(Zhu et al., 2024).

To this end, we propose, *Ego3D-VLM*, a training-free method that improves 3D spatial understanding of VLMs. The main idea of *Ego3D-VLM* is to create a **textual cognitive map** of the surrounding. The textual cognitive map defines a coordinate system center on the ego and locates important object (i.e., those referred to in the input prompt) in 3D coordinate space. Unlike point-clouds and BEV image methods (Team, 2025; Qi et al., 2024), our cognitive map only focuses on referred objects, making the number of input tokens significantly smaller and enabling efficient reasoning.

Given multi-view images as input, we first use referring expression comprehension (REC) models to find the 2D location of referred expressions in pixel space. We also use a metric depth estimator to estimate the depth values. We then convert the 2D points to 3D points in camera coordinate space and transform 3D points from all views to the global coordinate space (i.e., front camera

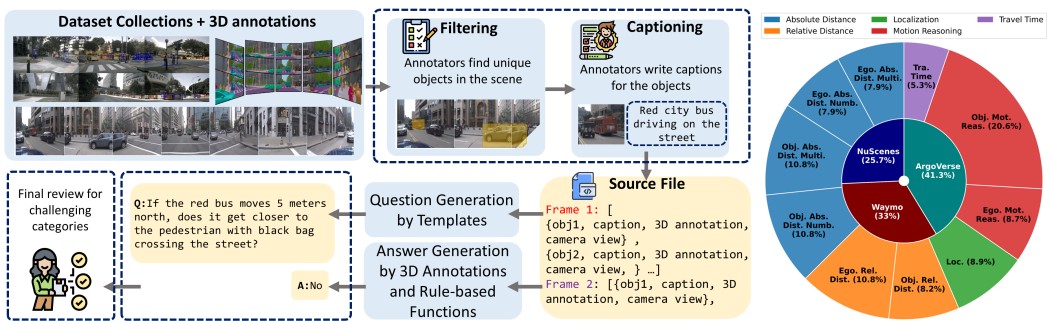

Figure 2: Overview of our `Ego3D-Bench` creation pipeline. Human annotators played a key role in the process. Right: the distribution of the samples. *Ego.*: Ego-centric, *Obj.*: Object-centric.

coordinate space). We define a cognitive map generator function that returns a textual cognitive map given 3D coordinates. The textual cognitive map defines a coordinate space for the VLM and organizes detected objects (expressions) based on the view-point. The textual cognitive map consistently improves SOTA VLMs on spatial reasoning. The major contributions of our work are:

- We introduce `Ego3D-Bench`, an ego-centric multi-view benchmark to evaluate 3D spatial understanding of VLMs.
- We propose *Ego3D-VLM*, a plug-and-play training-free framework to enhance 3D spatial understanding of VLMs, especially in ego-centric multi-view scenarios.
- Through extensive experiments, we demonstrate that *Ego3D-VLM* significantly improves SOTA generalist as well as 3D spatial VLMs in 3D reasoning.

## 2 RELATED WORK

**3D Spatial Benchmarks/Datasets for VLMs.** We categorize prior datasets and benchmarks into three groups based on the input data type. (1) (Liu et al., 2023; Kamath et al., 2023; Liao et al., 2024; Song et al., 2025; Zhang et al., 2025b; Cheng et al., 2024; Du et al., 2024) focus on single-view images. (2) (Yang et al., 2024; Daxberger et al., 2025; Li et al., 2025; Yang et al., 2025; Ray et al., 2025) focus on videos captured from indoor static scenes. As noted, this setup differs from the perceptual experience of real-world embodied agents, which involves ego-centric multi-view images. (3) All-Angle Yeh et al. (2025) is the first multi-view benchmark for VLMs. However, in their setup, multiple cameras observe a scene from different directions—similar to surveillance or motion capture systems. In contrast, `Ego3D-Bench` is the first ego-centric multi-view benchmark for VLMs created from dynamic outdoor scenes.

**3D Spatial VLMs.** These models—also referred to as 3D-LLMs or 3D-MLLMs—aim to perform tasks such as 3D grounding, spatial reasoning, depth estimation, and distance measurement. We categorize existing approaches into two main groups: (1) models that take point-clouds as input or reconstruct them from multi-view images, and (2) models that operate directly on image data.

(Hong et al., 2023b;a; Huang et al., 2024; Team, 2025; Zhang et al., 2025a; 2024a) fall in the first group. While point-cloud representations offer rich spatial information, they are difficult to reconstruct in dynamic scenes, often struggle with sparse multi-view data, and significantly increase inference time by over 10×. The 2nd group includes LLaVA-3D (Zhu et al., 2024), Video-3D LLM (Zheng et al., 2024), GPT4Scene (Qi et al., 2024), MM-Spatial (Daxberger et al., 2025), SpatialVLM (Chen et al., 2024), SpatialRGPT (Cheng et al., 2024), and SpatialPIN (Ma et al., 2024a). Our work falls in this image-based category.

LLaVA-3D and Video-3D LLM use depth maps and camera poses for 3D positional encoding. GPT4Scene leverages BEV to address 3D queries. However, these models are primarily trained on indoor, static scenes and limited in handling quantitative spatial relationships such as object distances. SpatialVLM addresses this limitation by introducing a synthetic data generation pipeline, enabling more robust spatial reasoning (remyxai, 2024). SpatialRGPT enhances input representation using region proposals alongside the original image. MM-Spatial proposes a VLM that supports

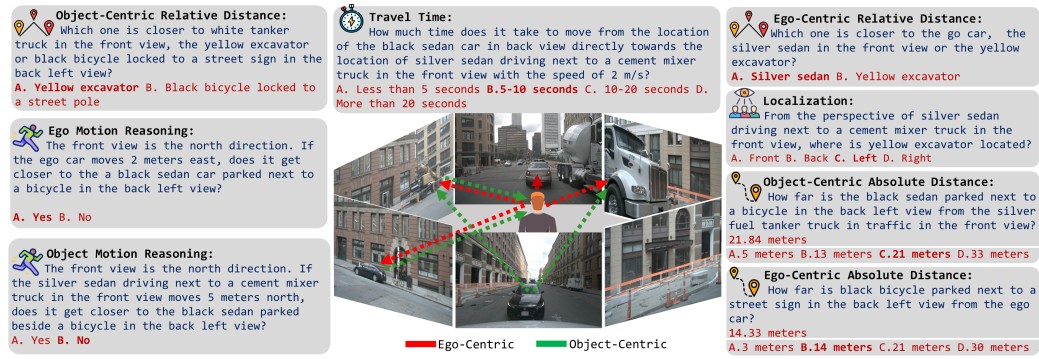

Figure 3: Ego- and object-centric samples from each category in `Ego3D-Bench`.

Chain-of-Thought spatial reasoning involving 2D grounding and depth estimation, and can also leverage depth input via tool-use. Different from SpatialRGPT, SpatialVLM, and MM-Spatial, our proposed *Ego3D-VLM* is a training-free method, can be applied to any existing VLM, and enhances spatial understanding of VLMs and outperforms prior works on ego-centric multi-view reasoning.

# 3 EGO3D-BENCH

`Ego3D-Bench` is designed to quantitatively evaluate 3D spatial understanding of VLMs from multi-view outdoor images. A major distinction between `Ego3D-Bench` and previous works is its focus on ego-centric multi-view data—an approach particularly relevant for applications in self-driving and robotics. `Ego3D-Bench` contains over 8.6K QA pairs in 5 distinct categories. This benchmark is constructed from the validation sets of 3 prominent outdoor multi-view datasets: nuScenes (Caesar et al., 2020), Waymo Open Dataset (Sun et al., 2020), and Argoverse 1 (Chang et al., 2019), featuring 6, 7, and 5 distinct camera views, respectively (Figure 2). These datasets cover diverse outdoor environments, including urban areas, highways, and rural regions. `Ego3D-Bench` leverages the multi-view nature of these datasets to formulate questions that require fine-grained visual comprehension across different viewpoints as well as reasoning over 3D spatial relationships. In this section, we describe the detailed process of constructing the questions of `Ego3D-Bench`.

## 3.1 BENCHMARK CONSTRUCTION

**Creation of Source Files:** Outdoor datasets, unlike their indoor counterparts, often feature multiple instances of the same object within each scene. While indoor environments typically contain unique items—such as a single oven or television per room—outdoor scenes commonly include numerous similar objects, like multiple cars or pedestrians. This makes it challenging to uniquely reference a target object in the scene. Thus, annotators begin by carefully reviewing each scene to identify unique objects (called "Filtering" in Figure 2). They then compose concise captions describing these objects (called "Captioning" in Figure 2). The descriptions are designed to be short, yet discriminative such that each object can be uniquely identified. Furthermore, the ground-truth 3D annotations (bounding boxes) are collected from the source datasets. Object captions, 3D annotations, and the camera view from which the object is visible are used to construct a source file for QA creation.

**Creation of QAs:** Each question category follows a predefined template with placeholders for object names, such as *How far is <obj1> in <view1> from <obj2> in <view2>?* (see Appendix A.3 for all question templates). Each question is constructed by placing the generated object annotation and camera views from the source file in the question template. To generate the answers, we use rule-based functions that leverage ground-truth 3D annotations. For challenging categories, such as motion reasoning, a final human review is conducted to ensure the accuracy of the QA pairs.

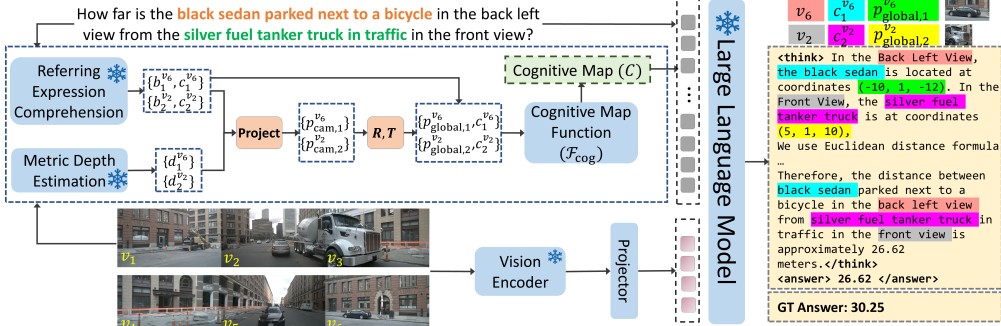

Figure 4: An overview of *Ego3D-VLM*, our training-free 3D spatial understanding framework.

## 3.2 BENCHMARK DETAILS

Figure 3 illustrates different question categories in Ego3D-Bench. We emphasize on questions that require understanding the 3D world by utilizing information from multiple views. Thus, we exclude questions that can be answered using a single-view or by analyzing each view independently (e.g., counting the number of objects), as they do not contribute to multi-view spatial reasoning. We define question from the perspective of both ego and objects in the scene. To clearly indicate the perspective of each question, we categorize them into ego-centric or object-centric. In the following, we describe the five tasks (each composed of ego-centric and object-centric types) in our benchmark.

**(1) Absolute Distance Measurement.** This category asks the VLM to estimate the absolute metric distance between the ego and another object in the scene or between two different objects from different views. This category is designed in two forms of multi-choice QA and absolute meter.

**(2) Relative Distance Measurement.** In this task, the VLM is asked to determine which of two objects is closer—either to the ego or to a specific object, designed in the form of two-choice QA.

**(3) Localization.** This category is only object-centric and assesses the VLM's ability to localize objects within a scene. Specifically, the model is asked to infer the location of object-1 from the perspective of object-2 in the form of multi-choice QA.

**(4) Motion Reasoning.** This category defines a coordinate system using cardinal directions and asks the VLM that if the ego or object-1 moves in a direction, whether or not it gets closer to or farther away from object-2. Answering this yes/no question requires mapping the spatial relationship between the objects and how the distances would change if one object moves in a specific direction.

**(5) Travel Time.** Given a specific motion speed, this category asks (via multi-choice QA) to estimate the required time to move from the location of either the ego or object-1 to the location of object-2.

**Evaluation Metrics.** We design most of the questions as multi-choice QA and use accuracy as the evaluation metric. We also have two absolute distance estimation for which we use root mean squared error (RMSE) in meters as the evaluation metric.

## 4 TRAINING-FREE 3D SPATIAL UNDERSTANDING : *Ego3D-VLM*

Figure 4 shows an overview of our proposed framework, *Ego3D-VLM*. Given a set of multi-view images and a natural language prompt, we use a REC model to detect all objects mentioned in the prompt. For each camera view $v \in \{1, 2, \dots, V\}$, the REC model returns a set of detected objects:

$$\mathcal{O}^{(v)} = \left\{ \left( b_i^{(v)}, c_i^{(v)} \right) \right\}_{i=1}^{N^{(v)}}, \tag{1}$$

where $b_i^{(v)} \in \mathbb{R}^4$ denotes the 2D bounding box coordinates for the $i$-th object in view $v$, and $c_i^{(v)}$ is the corresponding referring expression match. We compute the 2D pixel-space center of each bounding box as $\mathbf{u}_i^{(v)} \in \mathbb{R}^2$ to create a list of 2D centers of the objects in the pixel space.

Figure 5: Example question with associated textual and visual cognitive maps.

**Camera Coordinate Transformation.** To obtain 3D spatial info, we use a metric depth estimator to predict a dense depth map $D^{(v)} \in \mathbb{R}^{H \times W}$ for each view $v$. For each detected object center $\mathbf{u}_i^{(v)} = (x_i, y_i)^\top$, we extract the corresponding depth value $d_i^{(v)} = D^{(v)}(x_i, y_i)$. We then project each center point into the 3D camera coordinate system using the camera intrinsics $K^{(v)} \in \mathbb{R}^{3 \times 3}$:

$$\mathbf{p}_{\mathrm{cam},i}^{(v)} = d_i^{(v)} \cdot \left(K^{(v)}\right)^{-1} \begin{bmatrix} x_i \\ y_i \\ 1 \end{bmatrix} \in \mathbb{R}^3. \tag{2}$$

Next, we transform the 3D point from the local to the global coordinate system, i.e., defined as the front camera view point coordinate system. This replicates the human perception mechanism given multi-view cameras by using the front view as the reference building the 3D world based on that. Using the rotation matrix $R^{(v)} \in \mathbb{R}^{3 \times 3}$ and translation vector $T^{(v)} \in \mathbb{R}^{3 \times 3}$, we have:

$$\mathbf{p}_{\mathrm{global},i}^{(v)} = \begin{bmatrix} R^{(v)} & T^{(v)} \\ 0 & 1 \end{bmatrix} \cdot \begin{bmatrix} \mathbf{p}_{\mathrm{cam},i}^{(v)} \\ 1 \end{bmatrix} \in \mathbb{R}^4. \tag{3}$$

This process effectively simulates human spatial perception by leveraging multi-view images to construct a unified 3D representation of the scene.

**Relational Scaling.** Humans estimate object sizes using known references—e.g., knowing a person is $\approx 1.7$ m tall helps infer the size of nearby objects. Inspired by this, we scale 3D points $\mathbf{p}_{\mathrm{global},i}^{(v)}$ using familiar object categories (e.g., sedans, humans, bikes) identified in a few representative frames across all camera views. We compute the average observed height $h_{\mathrm{est}}$ and scale all 3D points by $s = h_{\mathrm{cs}}/h_{\mathrm{est}}$, where $h_{\mathrm{cs}}$ is the canonical common sense height (e.g., 1.7 m for humans). This yields $\mathbf{p}_{\mathrm{scaled},i}^{(v)} = s \cdot \mathbf{p}_{\mathrm{global},i}^{(v)}$, producing physically plausible scales without ground-truth depth.

**Creating a Cognitive Map.** We define a cognitive map generator function $\mathcal{F}_{\mathrm{cog}}$, which takes as input the set of 3D global coordinates and corresponding referring expressions for all detected objects across all camera views. Specifically, for each object $i$ detected from view $v$, we denote its global 3D position as $\mathbf{p}_{\mathrm{scaled},i}^{(v)}$ and its matched referring expression as $c_i^{(v)}$. The function outputs a textual cognitive map $C$, defined as:

$$C = \mathcal{F}_{\mathrm{cog}}\left(\left\{\left(\mathbf{p}_{\mathrm{scaled},i}^{(v)}, c_i^{(v)}\right)\right\}_{i,v}\right). \tag{4}$$

$\mathcal{F}_{\mathrm{cog}}$ constructs an ego-centric world model centered on the agent. It integrates multi-view detections and links each referred object to its spatial position and originating viewpoint. The resulting cognitive map captures both linguistic references and spatial semantics in a compact, human-interpretable form—enabling grounded reasoning and situational awareness. Figure 5 illustrates a sample prompt, the corresponding multi-view images, and the generated cognitive maps.

Given a VLM $\mathcal{V}$ that answers queries by visual-textual contexts, it takes as input the cognitive map $C$, a set of multi-view images $\mathcal{I} = \{I^{(v)}\}_v$, and a natural language query $q$, to return an answer $a$:

$$a = \mathcal{V}(C, \mathcal{I}, q). \tag{5}$$

The cognitive map $C$ provides structured spatial grounding, while $\mathcal{I}$ supplies rich visual cues—such as appearance, color, and fine-grained context—not captured in $C$. Together, they guide the VLM in interpreting and answering the query.

Table 1: Comparison results of generalist VLMs vs. *Ego3D-VLM* on `Ego3D-Bench`.

| Model | Ego Dist. | Obj. Dist. | Loc. | Ego Mot. | Obj. Mot. | Travel Time | Ego Rel. | Obj. Rel. | Avg.↑ | Ego Dist. | Obj. Dist. | Avg.↓ |
|---|---|---|---|---|---|---|---|---|---|---|---|---|
| | Accuracy (%) ↑ | | | | | | | | | RMSE↓ | | |
| Human Level | 78.2 | 57.1 | 100 | 100 | 95.3 | 72.7 | 85.0 | 94.1 | 85.3 | - | - | - |
| Chance Level | 25.8 | 25.5 | 24.3 | 49.7 | 50.6 | 24.0 | 49.9 | 50.1 | 37.5 | - | - | - |
| *Closed-source Models* | | | | | | | | | | | | |
| GPT-4o | 51.4 | 38.4 | 47.7 | 86.9 | 71.1 | 35.8 | 65.9 | 84.2 | 56.7 | 8.5 | 29.5 | 19.2 |
| *Ego3D-VLM* GPT-4o | 76.3 | 58.9 | 57.3 | 87.3 | 89.7 | 59.9 | 70.6 | 85.3 | **73.2** | 6.3 | 8.4 | **7.4** |
| Gemini-1.5-Pro | 51.3 | 35.8 | 49.4 | 88.4 | 70.6 | 37.7 | 55.5 | 71.4 | 57.5 | 10.7 | 28.6 | 19.6 |
| *Ego3D-VLM* Gemini-1.5-Pro | 65.0 | 62.0 | 67.2 | 93.9 | 91.4 | 58.4 | 66.2 | 80.6 | **73.1** | 6.6 | 7.8 | **7.2** |
| *Qwen2.5 Family* | | | | | | | | | | | | |
| Qwen2.5-3B | 21.5 | 29.4 | 28.8 | 50.3 | 41.9 | 30.9 | 54.1 | 56.1 | 39.1 | 30.5 | 33.7 | 32.1 |
| *Ego3D-VLM* Qwen2.5-3B | 35.0 | 30.0 | 29.8 | 52.4 | 56.4 | 29.6 | 60.1 | 62.0 | **44.4** | 12.7 | 12.5 | **12.6** |
| Qwen2.5-7B | 32.7 | 31.5 | 30.5 | 45.9 | 44.0 | 34.5 | 43.2 | 66.5 | 41.1 | 25.1 | 35.5 | 30.3 |
| *Ego3D-VLM* Qwen2.5-7B | 59.4 | 54.5 | 33.1 | 62.3 | 58.2 | 49.4 | 50.5 | 66.9 | **54.3** | 8.1 | 10.9 | **9.5** |
| Qwen2.5-32B | 45.4 | 40.7 | 49.6 | 75.6 | 74.1 | 40.1 | 54.0 | 79.0 | 57.3 | 21.2 | 10.4 | 15.8 |
| *Ego3D-VLM* Qwen2.5-32B | 63.7 | 62.6 | 54.5 | 87.7 | 86.2 | 40.8 | 62.6 | 72.0 | **65.5** | 11.6 | 15.9 | **13.7** |
| Qwen2.5-72B | 42.4 | 38.6 | 54.8 | 86.8 | 68.9 | 38.5 | 53.3 | 80.5 | 58.0 | 10.3 | 22.2 | 16.2 |
| *Ego3D-VLM* Qwen2.5-72B | 62.1 | 61.4 | 58.2 | 94.0 | 84.5 | 56.0 | 63.1 | 76.6 | **69.5** | 6.8 | 8.3 | **7.5** |
| *InternVL3 Family* | | | | | | | | | | | | |
| InternVL3-8B | 25.8 | 28.7 | 29.8 | 54.1 | 54.8 | 36.1 | 49.9 | 65.2 | 43.1 | 15.2 | 39.1 | 27.2 |
| *Ego3D-VLM* InternVL3-8B | 65.4 | 56.1 | 37.0 | 73.0 | 71.4 | 49.0 | 63.5 | 66.0 | **60.1** | 6.8 | 9.0 | **8.0** |
| InternVL3-14B | 46.0 | 35.6 | 35.9 | 63.2 | 65.9 | 41.6 | 55.5 | 70.1 | 51.7 | 10.6 | 24.5 | 17.6 |
| *Ego3D-VLM* InternVL3-14B | 70.3 | 60.9 | 50.5 | 79.8 | 83.1 | 50.2 | 63.0 | 70.7 | **66.1** | 6.6 | 8.8 | **7.7** |
| InternVL3-38B | 35.4 | 31.0 | 39.4 | 66.6 | 64.9 | 38.0 | 61.0 | 77.3 | 51.7 | 8.6 | 42.2 | 25.4 |
| *Ego3D-VLM* InternVL3-38B | 55.1 | 64.5 | 53.4 | 87.2 | 88.9 | 51.9 | 66.6 | 76.5 | **68.0** | 8.2 | 8.7 | **8.5** |
| InternVL3-78B | 54.6 | 48.4 | 50.3 | 77.7 | 70.0 | 44.8 | 57.0 | 76.6 | 59.9 | 12.0 | 15.5 | 13.8 |
| *Ego3D-VLM* InternVL3-78B | 68.3 | 62.7 | 62.9 | 91.6 | 89.2 | 55.1 | 66.3 | 78.2 | **71.8** | 6.8 | 8.1 | **7.4** |
| *Ovis2 Family* | | | | | | | | | | | | |
| Ovis2-4B | 29.8 | 28.9 | 18.4 | 47.5 | 48.1 | 36.9 | 54.2 | 70.3 | 41.8 | 17.1 | 29.5 | 23.3 |
| *Ego3D-VLM* Ovis2-4B | 62.1 | 46.9 | 20.1 | 49.1 | 51.9 | 33.3 | 62.0 | 60.4 | **48.2** | 6.5 | 10.4 | **8.5** |
| Ovis2-8B | 25.6 | 28.6 | 31.1 | 45.3 | 51.4 | 31.4 | 50.7 | 68.2 | 41.5 | 11.5 | 30.2 | 20.8 |
| *Ego3D-VLM* Ovis2-8B | 64.9 | 54.9 | 33.5 | 57.2 | 57.1 | 46.1 | 65.1 | 71.0 | **56.2** | 6.0 | 9.5 | **7.8** |
| Ovis2-16B | 41.7 | 36.5 | 41.1 | 50.5 | 52.5 | 27.9 | 50.9 | 74.9 | 47.0 | 10.8 | 16.6 | 13.7 |
| *Ego3D-VLM* Ovis2-16B | 63.4 | 58.1 | 43.7 | 53.9 | 73.4 | 51.1 | 61.0 | 82.9 | **60.9** | 6.6 | 8.3 | **7.4** |

# 5 EVALUATION ON EGO3D-BENCH

In this section, we present a comprehensive evaluation of VLMs on `Ego3D-Bench`. We organize our experiments into 4 parts by benchmarking (1) generalist VLMs (i.e., models trained for general vision-language tasks), (2) 3D-VLMs (i.e., models trained specifically for 3D SU), (3) VLM$_{+Depth+REC}$ (i.e., generalist VLMs augmented with depth and REC tools), and (4) ablation studies. We use a fixed R1-style prompt (Guo et al., 2025) in the evaluations of all models (details in the Appendix A.9). We use Grounding-Dino-Base (Research, 2023) as the REC model and Depth-Anything-V2-Metric-Large (Contributors, 2024) as the metric depth estimator in all experiments.

**Chance/Human Performance Levels.** We provide frequency-based random selection as the chance level baseline for the multi-choice questions. Furthermore, we conduct human evaluation on multi-choice questions, where 10% of the questions from each category are randomly sampled and evaluated by each human annotator. Table 1 presents the results of this analysis. While humans can accurately answer the questions that require reasoning about relative location of the objects in space, their performance degrades in questions that require estimation of the exact distance between objects. This highlights the challenging nature of accurate distance estimation.

## 5.1 BENCHMARKING GENERALIST VLMS

We use GPT-4o (OpenAI, 2024) and Gemini-2-Flash (DeepMind, 2023) closed-source VLMs, and 3 competitive families of open-source models: InternVL3 (et al., 2025), Qwen2.5-VL (Qwen et al.,

2025), and Ovis2 (Lu et al., 2024) open-source models. Inference time analysis, numerical results on more generalist VLMs, and qualitative examples are given in the Appendix A.6, A.5, and A.9.

As given in Table 1, the performance of the VLMs varies considerably across different model parameter sizes. Smaller models (e.g., 3B and 8B) operate near chance level, indicating limited capacity for multi-view 3D reasoning. In contrast, larger models demonstrate substantial improvements over the chance level, yet still exhibit a noticeable gap when compared to human level performance. Furthermore, our proposed *Ego3D-VLM* provides significant improvement across all model sizes and tasks (average 56% relative improvement in RMSE and 12% in Accuracy), underscoring the importance of providing structured spatial representations to the model in 3D understanding tasks.

**Performance Analysis.** Figure 6 shows the average performance of 4 leading models (GPT-4o, Gemini-1.5-Pro, InternVL3-78B, and Qwen2.5-VL-72B) with and without *Ego3D-VLM* integration across all multiple-choice QAs. *Travel time*, *localization*, and *object-centric absolute distance* are the most challenging tasks for VLMs with 40%-45% average accuracy. We argue that these categories require intricate spatial reasoning and the ability to construct an accurate mental map by the relative positioning of objects. While humans can simply build such maps and achieve perfect accuracy, VLMs struggle to replicate this level of SU, indicating a key area for further development. In the absolute distance tasks, the incorporation of 3D location data through the cognitive map substantially narrows the gap between model and human level accuracy. Notably, for object-centric absolute distance, VLMs augmented with *Ego3D-VLM* even surpass human. This is expected, as human estimation of 3D distances in object-centric cases is prone to substantial error without explicit 3D spatial info.

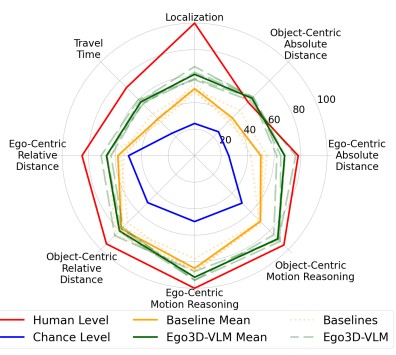

Figure 6: Average performance of leading VLMs w/ and w/o *Ego3D-VLM* vs. chance & human levels on each category of Ego3D-Bench.

Conversely, in the localization task, VLMs continue to fall short of human proficiency, even with cognitive map support.

**Blind VLM Performance.** This baseline evaluates how much spatial reasoning can be achieved by VLMs using only textual input, without any visual information, relying solely on their world knowledge (Majumdar et al., 2024). We report the average performance of GPT-4o and Gemini-1.5-Pro models. The blind VLMs perform 5% worse than vision-enabled VLMs (53.8% vs. 58.8%) and 16.4% better than chance level. These results are consistent with findings from prior single-view and video-based spatial reasoning benchmarks (Cheng et al., 2024; Yang et al., 2024).

**Blind *Ego3D-VLM* Performance.** In this section, we run an experiment by feeding the cognitive map to a text only LLM. Since InternVL3-8B/14B use Qwen2.5-7B/14B as their backbone LLM, we use the same models for fair comparison. Table 2 shows that Ego3D-VLM with VLM outperforms Ego3D-VLM with LLM since VLMs can ignore false positives in the cognitive map and remain robust when false negatives occur, whereas LLMs suffer performance drops in those cases.

## 5.2 BENCHMARKING 3D-VLMS

3D-VLM models have been trained on datasets designed for 3D SU, such as absolute distance estimation and relative location inference. We benchmark Spatial-RGPT and two checkpoints trained with the Spatial-VLM framework (Chen et al., 2024): SpaceThinker-Qwen2.5-3B (remyxai, 2024) and SpaceQwen2.5-3B. The Spatial-RGPT (Cheng et al., 2024) model assumes that

Table 2: Blind *Ego3D-VLM* performance analysis.

| Model | Only LLM | Mult. Choice (Acc. ↑) | Abs. Ans. (RMSE↓) |
|---|---|---|---|
| InternVL3-8B | | 43.1 | 27.2 |
| Ego3D-VLM$_{InternVL3-8B}$ | ✓ | 53.1 | 11.4 |
| Ego3D-VLM$_{InternVL3-8B}$ | | **60.1** | **8.0** |
| InternVL3-14B | | 51.7 | 17.6 |
| Ego3D-VLM$_{InternVL3-14B}$ | ✓ | 55.6 | 10.8 |
| Ego3D-VLM$_{InternVL3-14B}$ | | **68.0** | **8.5** |

specific regions of the image are annotated with bounding boxes and is trained to answer 3D questions based on these regions. To evaluate SpatialRGPT on our Ego3D-Bench, we reformat the input: object names in the questions are replaced with placeholder labels (e.g., *region-i*), and a list

of corresponding region captions is passed to the REC model to generate bounding boxes. These estimated bounding boxes were then overlaid on the images before being fed to SpatialRGPT.

Table 3 presents the performance of the above-mentioned 3D-VLMs on our `Ego3D-Bench`. SpaceThinker-Qwen2.5-3B achieves the highest overall performance, surpassing both SpatialRGPT with 8B parameters and other generalist VLMs of similar scale. This outcome highlights the critical role of dedicated 3D spatial pretraining and end-to-end architecture in advancing VLM capabilities for spatial reasoning tasks. Moreover, augmenting SpaceThinker with *Ego3D-VLM* leads to an average improvement of 3% on multiple-choice questions and a reduction of more than 4 meters in absolute distance RMSE, emphasizing the effectiveness of our proposed solution.

Table 3: Comparison results of 3D-VLMs and our method (*Ego3D-VLM*) on `Ego3D-Bench`.

| Model | Accuracy (%) ↑ | | | | | | | | | RMSE↓ | | |
| --- | --- | --- | --- | --- | --- | --- | --- | --- | --- | --- | --- | --- |
| | Ego Dist. | Obj. Dist. | Loc. | Ego Mot. | Obj. Mot. | Travel Time | Ego Rel. | Obj. Rel. | **Avg.↑** | Ego Dist. | Obj. Dist. | **Avg.↓** |
| SpaceQwen2.5-VL-3B | 18.1 | 21.4 | 16.1 | 35.6 | 35.2 | 30.2 | 31.2 | 32.0 | 27.5 | 10.6 | 15.7 | 13.2 |
| SRGPT-VILA1.5-8B | 45.7 | 35.5 | 39.6 | 43.6 | 45.8 | 24.9 | 50.8 | 71.7 | 44.7 | 11.5 | 15.1 | 13.3 |
| SThinker-Qwen2.5-3B | 38.9 | 39.0 | 21.9 | 57.5 | 53.4 | 27.1 | 52.8 | 71.6 | 45.2 | 15.2 | 16.9 | 16.0 |
| *Ego3D-VLM*SThinker-Qwen2.5-3B | 50.6 | 44.2 | 26.4 | 53.4 | 57.0 | 42.6 | 54.7 | 59.8 | **48.6** | 11.1 | 12.2 | **11.6** |

## 5.3 BENCHMARKING VLM$_{+\text{DEPTH}+\text{REC}}$.

This category enhances the base VLM with a metric depth estimator and a REC model. We pass each query to the REC model (Research, 2023) to extract BBox of the referred objects, and use a metric depth estimator (Contributors, 2024) to estimate depth values. We then construct a list that pairs each matched referring expression with its corresponding BBox and estimated depth (i.e., distance from the ego). An example entry in this list is:
`[Front-View: Detected pedestrian with red hat at bbox [x1, y1, x2, y2], depth: z, Back-View:...]`
(examples in the Appendix A.9). As shown in Table 4, equipping the VLM with REC and depth estimator models improves its baseline performance, highlighting the base model's limitations in accurately identifying objects and estimating their depths. However, even with these enhancements, it falls short compared to our proposed *Ego3D-VLM* framework. This underscores the importance of integrating depth information into a unified map representation, enabling the VLM to reason more effectively about the 3D relationships between objects across different views.

Table 4: *Ego3D-VLM* vs. generalist VLMs with REC+depth tools on `Ego3D-Bench`.

| Model | Mult. Choice (Acc. ↑) | Abs. Ans. (RMSE↓) |
| --- | --- | --- |
| InternVL3-8B | 43.1 | 27.2 |
| InternVL3-8B$_{+\text{Depth}+\text{REC}}$ | 51.6 | 13.1 |
| *Ego3D-VLM* InternVL3-8B | **60.1** | **8.0** |
| Qwen2.5-7B | 41.1 | 30.3 |
| Qwen2.5-7B$_{+\text{Depth}+\text{REC}}$ | 49.4 | 11.8 |
| *Ego3D-VLM* Qwen2.5-7B | **54.3** | **9.5** |

## 5.4 EGO3D-VLM ON FURTHER BENCHMARKS

Our *Ego3D-VLM* is designed for ego-centric multi-view cases, which are particularly relevant for AI agent applications. In this section, however, we evaluate the performance of *Ego3D-VLM* in alternative multi-view settings, specifically those used in All-Angle Bench (Yeh et al., 2025) and VSI-Bench (Yang et al., 2024). Table 5 presents the performance of *Ego3D-VLM* compared to open- and closed-source baseline VLMs on All-Angle Bench and VSI-Bench. Despite the

Table 5: Comparison results on All-Angle Bench and VSI-Bench (accuracy).

| Model | All-Angle-Bench | VSI-Bench |
| --- | --- | --- |
| GPT-4o | 47.8 | 34.0 |
| Gemini-1.5-Pro | 47.4 | 45.4 |
| Gemini-1.5-Flash | 46.6 | 42.1 |
| InternVL3-8B | 47.9 | 38.1 |
| *Ego3D-VLM*InternVL3-8B | **49.5** | **39.6** |
| InternVL3-14B | 50.3 | 38.2 |
| *Ego3D-VLM*InternVL3-14B | **52.1** | **40.0** |

differences in data settings compared to our primary benchmark, *Ego3D-VLM* still outperforms the respective baselines, demonstrating its adaptability across diverse multi-view scenarios. More details are given in the Appendix A.6.

## 5.5 ABLATION STUDIES

***Ego3D-VLM* Components.** Table 6 presents an ablation on *Ego3D-VLM* core components. Starting from the baseline, $v_0$, we incrementally add each component. In $v_1$, we use a cognitive map

with estimated rotation ($R$), translation ($T$), and intrinsic parameters ($K$). Specifically, all cameras are positioned at the coordinate center with $T = [0,0,0]$. The front camera uses an identity rotation matrix, while all the other cameras—front-right, right, back-right, back, etc. —are incrementally rotated 45° around the Y-axis. The focal length is estimated from the images' approximate field of view. In $v_2$, we use the actual $K$ and in $v_3$ we further use actual $R$, and $T$ relative to the front camera, i.e., often available as a fixed parameter for an embodied AI agent. $v_3$ and $v_1$ obtain a comparable RMSE, while $v_3$ is only 2.4% higher in multi-choice QAs. Thus, even with estimated camera parameters our cognitive map can significantly boost the baseline's performance.

$v_4$ adds relational scaling to cognitive map which decreases the RMSE by 2.5 meters. $v_4$ is indeed *Ego3D-VLM* with all components. We provide two more ablations to evaluate the upper-bound of *Ego3D-VLM*. $v_5$ assumes that the list of objects (only their names) in the input query are provided which enhances the REC results. $v_6$ is the ground-truth cognitive map with ground-truth 3D locations. The difference between $v_6$ and human level is only 5% showing the upper bound of *Ego3D-VLM*.

Table 6: Ablation on *Ego3D-VLM* main components. $v_4$: all components.

| | Mult. Choice (Acc. ↑) | Abs. Ans. (RMSE↓) |
|---|---|---|
| $v_0$ InternVL3-8B | 43.1 | 27.2 |
| $v_1$   + CogMap (est. $R, T, K$) | 56.0 | 10.8 |
| $v_2$   + $K$ | 56.3 | 10.1 |
| $v_3$   + $R, T$ | 58.4 | 10.4 |
| $v_4$   + Scaling | 60.1 | 8.0 |
| $v_5$   + List of objects | 61.8 | 6.5 |
| $v_6$   GT CogMap | 79.4 | 1.3 |

**Perception-Reasoning Disentanglement.** In order to disentangle perception from reasoning, we add the ground-truth 2D BBox for all objects to the benchmark. This allows repeating our experiments with BBox overlaid, effectively isolating the reasoning and perception. As expected, adding GT BBox as visual prompts to the images will help the VLM to handle perception easier, therefore improving the results. The results in Table 7 show that the major limitation of baselines

Table 7: Perception-reasoning disentanglement.

| Model | GT BBOX | Mult. Choice (Acc. ↑) | Abs. Ans. (RMSE↓) |
|---|---|---|---|
| InternVL3-8B | | 43.1 | 27.2 |
| InternVL3-8B | ✓ | **50.2** | **21.0** |
| Ego3D-VLM$_{\text{InternVL3-8B}}$ | | 60.1 | 8.0 |
| Ego3D-VLM$_{\text{InternVL3-8B}}$ | ✓ | **62.2** | **6.8** |
| InternVL3-14B | | 51.7 | 17.6 |
| InternVL3-14B | ✓ | **51.8** | **16.2** |
| Ego3D-VLM$_{\text{InternVL3-14B}}$ | | 66.5 | 7.7 |
| Ego3D-VLM$_{\text{InternVL3-14B}}$ | ✓ | **66.8** | **7.5** |

is not 2D perception, but 3D perception and reasoning. More reasoning analysis on `Ego3D-Bench` as well as *Ego3D-VLM*'s robustness in challenging conditions are given in Appendix A.4 and A.7.

**Cognitive Map Format.** We explore 3 different formats for our generated cognitive map: visual, JSON, and textual. Examples of the visual and textual maps are shown in Figure 5. The average results are presented in Table 8. Among the three, the textual and JSON formats achieve the best overall performance.

Table 8: Ablation on cognitive map format.

| | Mult. Choice (Acc. ↑) | Abs. Ans. (RMSE↓) |
|---|---|---|
| Visual Cog-Map | 50.9 | 14.4 |
| JSON Cog-Map | 60.0 | 8.4 |
| Textual Cog-Map | **60.1** | **8.0** |

## 6 CONCLUSION AND FUTURE WORK

In this work, we proposed `Ego3D-Bench`, a benchmark for spatial understanding of VLMs on egocentric multi-view images. Overall, the benchmark shows a significant gap between human scores and VLMs. To address this limitation, we provided a training-free solution named *Ego3D-VLM* to enhance the performance of VLMs. Future work should explore fine-tuning VLMs with ego-centric multi-view QAs and incorporating 3D projection modules proposed in *Ego3D-VLM* in the course of fine-tuning. The limitations of this work are provided in the Appendix (A.10).

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

# A APPENDIX

In this appendix, we present the following additional discussions and experimental results corresponding to our proposed `Ego3D-Bench` and *Ego3D-VLM*:

- Code and `Ego3D-Bench`
- `Ego3D-Bench` vs. Other Benchmarks
- Further details on `Ego3D-Bench` creation and question templates
- Benchmarking the thinking of `Ego3D-Bench` as an Open-QA
- Results of the three source datasets: NuScenes, Waymo, and Argoverse
- Benchmarking more generalist VLMs
- Inference time analysis of *Ego3D-VLM* vs. other baselines
- Robustness of *Ego3D-VLM* in challenging conditions
- Qualitative results and prompts structure
- Limitations and future work

## A.1 CODE

In order for our results to be reproducible, we share our code as supplementary materials, with detailed instructions included in the associated README.md file. We have also included the `Ego3D-Bench` in the supplementary material. However, due to size limitation, images should be downloaded from the source datasets. Instructions to download images and models are provided in the README.md file.

## A.2 EGO3D-BENCH VS. OTHER BENCHMARKS

As described in the main body of this paper, our proposed *Ego3D-VLM* is designed for ego-centric multi-view scenarios, which are particularly relevant for AI agent applications. *Ego3D-VLM* is different from alternative multi-view settings, specifically those used in All-Angle Bench (Yeh et al., 2025) and VSI-Bench (Yang et al., 2024). Figure 7 illustrates the configurations of these benchmarks in comparison to `Ego3D-Bench`. All-Angle Bench features stationary multi-view cameras observing the same scene from different angles—a setup commonly used in motion capture and surveillance systems. VSI-Bench, on the other hand, involves a single moving camera within a static indoor environment, typically used for static scene reconstruction and SU. In contrast, our `Ego3D-Bench` assumes multiple cameras mounted on a moving ego agent, capturing the surrounding environment—a configuration suited for AI agent applications such as robotics and autonomous driving.

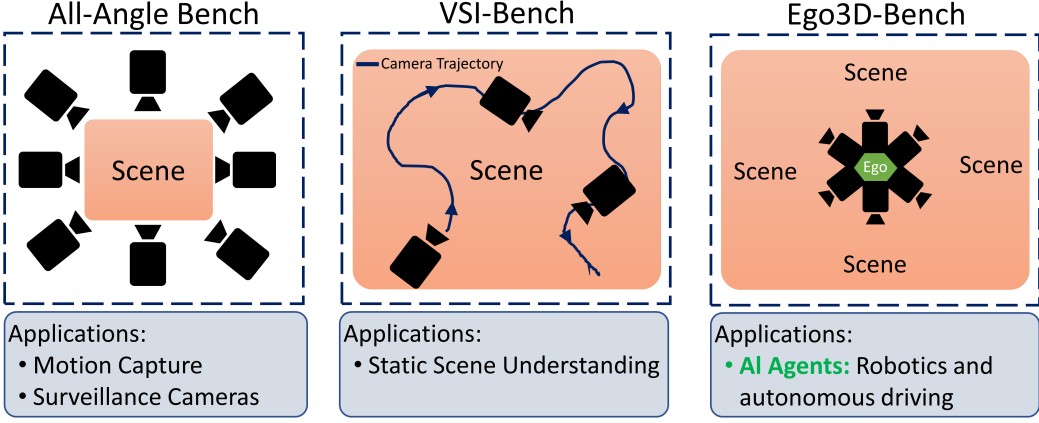

Figure 7: Settings of multi-view/video spatial understanding benchmarks for VLMs.

### A.3 EGO3D-BENCH: FURTHER DETAILS

In this section, we provide the templates used to create questions of the benchmark. Figure 8 shows templates used to create each category of `Ego3D-Bench`. We replace `<obj1>`, `<obj2>`, and `<obj3>` in the templates with object descriptions. `<view1>`, `<view2>`, and `<view3>` are replaced with the camera views from which the object is visible (e.g., Front-Right, Left, etc.). The placeholder `<direction>` in motion reasoning tasks is substituted with one of the four cardinal directions—"north", "east", "west", or "south"—and this process is repeated until all directions have been used. The placeholder `<Y>` is for motion reasoning tasks and is replaced with a number between 2 to 5 meters. `<X1>`, `<X2>`, `<X3>`, and `<X4>` serve as placeholders for multiple-choice options in absolute distance estimation tasks. One of these options is replaced with the ground-truth distance, while the remaining are filled with randomly generated values, ensuring that the distance between any two options is at least 8 meters.

### A.4 BENCHMARKING THE THINKING OF EGO3D-BENCH AS AN OPEN-QA

All our question-answer pairs include a "thinking" step in open-ended format. In this section, we generate "GT thinking" for further open-ended type evaluation. To this end, we used the GT cognitive maps along with GPT-4o to generate the reasoning for two categories that require numerical answers: Ego-Centric / Object-Centric Absolute Distance (in meters). Table 9 shows that Ego3D-VLM thinking is better than the baselines.

Table 9: Benchmarking the thinking of VLMs on `Ego3D-Bench` as an open-ended category. The GT thinking is generated using GT cognitive maps along with GPT-4o.

| | Thinking of Obj. Abs. Dist. GPT-4o Score (0-10)↑ | Thinking of Ego Abs. Dist. GPT-4o Score (0-10)↑ |
|---|---|---|
| InternVL3-8B | 1.9 | 2.0 |
| Ego3D-VLM $_{\text{InternVL3-8B}}$ | **4.7** | **5.2** |
| InternVL3-14B | 2.3 | 2.4 |
| Ego3D-VLM $_{\text{InternVL3-14B}}$ | **5.7** | **5.0** |
| InternVL3-38B | 2.1 | 2.5 |
| Ego3D-VLM $_{\text{InternVL3-38B}}$ | **7.1** | **3.7** |
| InternVL3-78B | 2.1 | 1.7 |
| Ego3D-VLM $_{\text{InternVL3-78B}}$ | **7.0** | **6.8** |

### A.5 BENCHMARKING GENERALIST VLMS: FURTHER RESULTS

In the main paper, we reported results for InternVL3, Qwen2.5-VL, and Ovis-2 as representative open-source models, as they are, to the best of our knowledge, the current SOTA among open-source VLMs. Table 10 extends this comparison by including additional generalist VLMs on `Ego3D-Bench`, such as Gemini-1.5-Flash (DeepMind, 2023), Gemini-2-Flash (DeepMind, 2023), Gemini-2.5-Flash (DeepMind, 2023), Phi-3.5 (et al., 2024), LLaVA-One-Vision-7B (Li et al., 2024a), LLaVA-Next-Video-7B (Zhang et al., 2024b), and DeepSeek-VL2 (Wu et al., 2024c). We observe that LLaVA-One-Vision, Phi-3.5, and DeepSeek-VL2 underperform compared to InternVL3 and Qwen2.5-VL models of similar size. For instance, InternVL3-8B achieves an accuracy of 43%, while LLaVA-One-Vision-7B achieves 38.7%. Likewise, Phi-3.5 (3.8B) attains an accuracy of 40.3%, compared to 41% from Ovis-2 (4B).

### A.6 INFERENCE TIME ANALYSIS

Table 11 shows the inference time analysis of *Ego3D-VLM* compared to baseline models. We report End-to-End Latency (E2E Lat.) in seconds and Peak Memory in GB. For the E2E Lat., we measure the average end-to-end inference of models on 50 samples of `Ego3D-Bench` and for the peak memory we report the peak memory usage during inference on the same samples. Experiments are performed using flash-attention-2. The memory and latency overhead of *Ego3D-VLM* over InternVL3-78B is 0.6% and 31%, respectively. The main reason for the latency overhead is that the model reasons more when cognitive map is provided in the *Ego3D-VLM*. In order to make the inference more efficient and deal with the latency overhead of the VLMs, post-training tech-

❑ **Travel Time:**
How much time does it take to move from the location of <obj1> in <view1> directly
towards the location of <obj2> in the <view2> with the speed of {V} m/s?
A.Less than 5 seconds B.5-10 seconds C. 10-20 seconds D. More than 20 seconds

❑ **Localization:**
- **Type1:**
    The front-view corresponds to the north direction. If I stand at the location of
    <obj1> in the <view1> facing north, is the <obj2> to my left, right, front, or
    back?
    A.Front  B. Back  C. Right  D. Left
- **Type2:**
    The front-view corresponds to the north direction. Assume <obj1> in the <view1> is
    facing north. From the perspective of <obj1> in the <view1>, where is <obj2> in
    the <view2> located?
    A. Front  B. Back  C. Right  D. Left

❑ **Ego Motion Reasoning:**
The front view corresponds to the north direction. If the ego car moves <Y> meters
{direction} while all other objects remain stationary, does the ego car get closer to
the <obj1> in the <view1>?
A. Yes B. No

❑ **Object Motion Reasoning:**
The front view corresponds to the north direction. If the <obj1> in the <view1> moves
<Y> meters <direction> while all other objects remain stationary, does the <obj1> in
the <view-1> get closer to the <obj2> in the <view2>?
A. Yes B. No

❑ **Object-Centric Relative Distance:**
Which one is closer to <obj1> in the <view1>, <obj2> in the <view2> or <obj3> in the
<view3>?
A. <obj2> B. <obj3>

❑ **Ego-Centric Relative Distance:**
Is <obj1> in the <view1> closer to the ego car than <obj2> in the <view2>?
A. Yes B. No

❑ **Ego-Centric Absolute Distance (Meter):**
How far, in meters, is <obj1> in the <view1> from the ego car?

❑ **Object-Centric Absolute Distance (Meter):**
How far, in meters, is <obj1> in the <view1> from <obj2> in the <view2>?

❑ **Ego-Centric Absolute Distance (Multi-choice):**
How far, in meters, is <obj1> in the <view1> from the ego car?
A. <X1> meters B. <X2> meters C. <X3> meters D. <X4> meters

❑ **Object-Centric Absolute Distance (Multi-Choice):**
How far, in meters, is <obj1> in the <view1> from <obj2> in the <view2>?
A. <X1> meters B. <X2> meters C. <X3> meters D. <X4> meters

Figure 8: Templates used to create questions of `Ego3D-Bench`.

niques such as quantization (Gholami et al., 2025), token pruning (Alvar et al., 2025), or knowledge
distillation (Gholami et al., 2024a) can be used.

## A.7  ROBUSTNESS OF *Ego3D-VLM* IN CHALLENGING CONDITIONS

To isolate tool performance, we included an ablation with ground-truth cognitive maps in the main
body of the paper (Table 6), which improve the results from 60.1% to 79.4% accuracy. This confirms

Table 10: Results of further generalist VLMs on `Ego3D-Bench`.

| | Accuracy (%) ↑ | | | | | | | | | RMSE↓ | | |
|---|---|---|---|---|---|---|---|---|---|---|---|---|
| Model | Ego Dist. | Obj. Dist. | Loc. | Ego Mot. | Obj. Mot. | Travel Time | Ego Rel. | Obj. Rel. | **Avg.↑** | Ego Dist. | Obj. Dist. | **Avg.↓** |
| *Gemini* | | | | | | | | | | | | |
| Gemini-1.5 Flash | 40.4 | 26.9 | 50.5 | 70.1 | 61.8 | 28.0 | 53.6 | 75.6 | 50.9 | 10.8 | 31.0 | 20.9 |
| Gemini-2-Flash | 43.3 | 38.6 | 54.1 | 58.0 | 39.3 | 29.1 | 47.1 | 68.5 | 47.2 | 10.9 | 29.4 | 20.1 |
| Gemini-2.5-Flash | 41.3 | 25.6 | 65.1 | 93.4 | 84.8 | 20.1 | 63.5 | 68.5 | 57.8 | 11.4 | 19.6 | 15.5 |
| *Phi* | | | | | | | | | | | | |
| Phi3.5 | 28.3 | 30.8 | 24.2 | 59.6 | 55.9 | 21.3 | 45.3 | 56.7 | 40.3 | 22.8 | 49.4 | 36.1 |
| *LLaVA Family* | | | | | | | | | | | | |
| LLaVA-Next-Video-7B | 30.2 | 26.6 | 42.2 | 60.8 | 58.0 | 48.3 | 55.1 | 73.5 | 49.3 | 19.5 | 22.8 | 21.2 |
| LLaVA-OV-7B | 23.5 | 22.1 | 58.4 | 56.3 | 52.5 | 17.9 | 59.0 | 20.0 | 38.7 | 17.8 | 19.3 | 18.5 |
| *DeepSeek Family* | | | | | | | | | | | | |
| DeepSeek-VL2-tiny | 34.7 | 29.0 | 19.9 | 60.0 | 59.3 | 46.6 | 57.8 | 65.0 | 46.5 | 14.0 | 18.7 | 16.3 |
| DeepSeek-VL2-Small | 18.1 | 19.2 | 28.6 | 50.3 | 55.6 | 37.4 | 50.5 | 59.3 | 39.9 | 12.7 | 14.7 | 13.7 |
| DeepSeek-VL2 | 22.3 | 25.3 | 32.4 | 60.0 | 59.1 | 21.4 | 56.7 | 62.0 | 42.4 | 20.3 | 17.4 | 18.8 |

Table 11: Inference time and memory usage of *Ego3D-VLM* compared to the baselines.

| | E2E Lat. (sec) | Memory (GB) | | E2E Lat. (sec) | Memory (GB) |
|---|---|---|---|---|---|
| InternVL3-8B | 5.2 | 18.1 | InternVL3-38B | 16.9 | 80.0 |
| *Ego3D-VLM*InternVL3-8B | 8.6 | 26.5 | *Ego3D-VLM*InternVL3-38B | 19.1 | 84.6 |
| InternVL3-14B | 15.5 | 33.1 | InternVL3-78B | 35.0 | 161.7 |
| *Ego3D-VLM*InternVL3-14B | 16.4 | 40.2 | *Ego3D-VLM*InternVL3-78B | 46.9 | 162.4 |

that, there is still a gap between the accuracy of such imperfect tools and ground-truth REC/depth. However, compared to the baseline, our solution still achieves more robust results even in challenging conditions such as low brightness, motion blur, and occlusion. To support this claim, we simulated these conditions and re-ran more specific experiments shown in the Table 12.

Although G-DINO and Depth-Anything used in Ego3D-VLM are among the best tools for REC and depth estimation, our solution is orthogonal to any such tools, and has no limitation in this regard.

Table 12: Robustness of external tools in scenarios involving occlusion, motion blur, and low light.

| | **Mult. Choice** (Acc. ↑) | **Abs. Ans.** (RMSE↓) |
|---|---|---|
| InternVL3-8B | 43.1 | 27.2 |
| Ego3D-VLM InternVL3-8B | **60.1** | **8.0** |
| InternVL3-8B [60% Darkness] | 41.1 | 29.8 |
| Ego3D-VLM InternVL3-8B [60% Darkness] | **59.6** | **10.6** |
| InternVL3-8B [Motion Blur, 15×1 kernel] | 42.5 | 28.5 |
| Ego3D-VLM InternVL3-8B [motion blur, 15×1 kernel] | **57.9** | **9.9** |
| InternVL3-8B [30% Occlusion] | 42.0 | 28.9 |
| Ego3D-VLM InternVL3-8B [30% Occlusion] | **58.7** | **10.7** |

## A.8 RESULTS OF DIFFERENT SOURCE DATASETS

Table 13 presents the results for different source dataset splits used to create `Ego3D-Bench`. Despite the varying number of camera viewpoints across the three datasets, the performance deviations are minimal. This consistency underscores the reliability of `Ego3D-Bench` as a benchmark, indicating that model performance is not heavily influenced by the specific choice of source data—a desirable property for robust and fair benchmark design.

## A.9 QUALITATIVE RESULTS AND PROMPTS STRUCTURE

Figures 10-20 demonstrate example responses of InternVL3-78B and *Ego3D-VLM*InternVL3-78B on all categories of `Ego3D-Bench`. As seen, `Ego3D-Bench` enhances the spatial reasoning ability of the baseline by providing the textual cognitive map.

Table 13: Results on samples of `Ego3D-Bench` generated from each of the source datasets.

| | NuScenes | | Waymo | | Argoverse | |
|---|---|---|---|---|---|---|
| | Acc↑ | RMSE↓ | Acc↑ | RMSE↓ | Acc↑ | RMSE↓ |
| *Closed-source Models* | | | | | | |
| GPT-4o | 60.9 | 21.2 | 60.0 | 19.0 | 59.4 | 17.1 |
| *Ego3D-VLM* $_{\text{GPT-4o}}$ | **73.5** | **8.5** | **73.1** | **7.5** | **72.7** | **6.2** |
| Gemini-1.5-Pro | 59.3 | 26.4 | 55.6 | 12.7 | 57.5 | 19.5 |
| *Ego3D-VLM* $_{\text{Gemini-1.5-Pro}}$ | **77.1** | **8.1** | **68.9** | **6.3** | **73.0** | **7.1** |
| *Qwen2.5 Family* | | | | | | |
| Qwen2.5-3B | 39.7 | 33.4 | 38.6 | 31.5 | 38.9 | 31.2 |
| *Ego3D-VLM* $_{\text{Qwen2.5-3B}}$ | **44.7** | **13.0** | **40.3** | **12.1** | **48.1** | **12.6** |
| Qwen2.5-7B | 40.9 | 36.0 | 40.6 | 24.7 | 41.6 | 30.1 |
| *Ego3D-VLM* $_{\text{Qwen2.5-7B}}$ | **56.4** | **11.3** | **51.9** | **8.3** | **54.4** | **8.7** |
| Qwen2.5-32B | 56.6 | 17.5 | 56.0 | 11.0 | 59.2 | 18.7 |
| *Ego3D-VLM* $_{\text{Qwen2.5-32B}}$ | **64.3** | **17.1** | **64.8** | **10.6** | **64.0** | **12.1** |
| Qwen2.5-72B | 58.2 | 21.2 | 57.7 | 11.1 | 58.2 | 16.2 |
| *Ego3D-VLM* $_{\text{Qwen2.5-72B}}$ | **74.4** | **8.5** | **64.4** | **6.4** | **69.5** | **7.6** |
| *InternVL3 Family* | | | | | | |
| InternVL3-8B | 42.3 | 28.3 | 44.0 | 21.7 | 42.9 | 31.5 |
| *Ego3D-VLM* $_{\text{InternVL3-8B}}$ | **62.0** | **10.5** | **57.1** | **6.5** | **61.3** | **6.7** |
| InternVL3-14B | 51.7 | 20.4 | 50.2 | 15.8 | 53.3 | 16.6 |
| *Ego3D-VLM* $_{\text{InternVL3-14B}}$ | **69.0** | **8.8** | **63.7** | **7.5** | **67.0** | **6.3** |
| InternVL3-38B | 51.2 | 31.4 | 50.5 | 19.4 | 53.4 | 25.4 |
| *Ego3D-VLM* $_{\text{InternVL3-38B}}$ | **71.7** | **8.7** | **64.3** | **8.2** | **67.0** | **6.3** |
| InternVL3-78B | 60.0 | 16.7 | 59.7 | 10.7 | 60.1 | 13.7 |
| *Ego3D-VLM* $_{\text{InternVL3-78B}}$ | **76.5** | **8.3** | **67.0** | **6.5** | **71.0** | **7.5** |
| *Ovis2 Family* | | | | | | |
| Ovis2-4B | 41.2 | 29.2 | 43.0 | 18.1 | 40.9 | 22.5 |
| *Ego3D-VLM* $_{\text{Ovis2-4B}}$ | **47.3** | **9.8** | **47.5** | **7.1** | **49.7** | **8.3** |
| Ovis2-8B | 41.5 | 21.4 | 41.9 | 20.7 | 41.0 | 20.2 |
| *Ego3D-VLM* $_{\text{Ovis2-8B}}$ | **55.6** | **9.4** | **55.1** | **6.8** | **57.9** | **7.0** |
| Ovis2-16B | 48.6 | 16.2 | 45.5 | 11.9 | 46.8 | 12.9 |
| *Ego3D-VLM* $_{\text{Ovis2-16B}}$ | **62.6** | **8.5** | **57.9** | **7.15** | **62.2** | **6.5** |

## A.10 LIMITATIONS AND FUTURE WORK

This work has several limitations that need further investigation. Ego3D-VLM does not improve the innate visual spatial reasoning of VLMs, but rather proposes a framework that augments VLMs with structured 3D information. Future work, should consider end-to-end training of Ego3D-VLM using reinforcement learning Rezaei et al. (2026); Shao et al. (2024) to enhance the spatial perception capability of VLMs. The proposed *Ego3D-VLM* relies on the reasoning capabilities of the underlying vision-language models (VLMs). Consequently, for models with limited reasoning ability (e.g., VILA1.5-8B), we observe little to no improvement when combined with *Ego3D-VLM*. In this work, we did not address spatial reasoning tasks in conjunction with temporal understanding. Future work should extend the proposed method to incorporate the temporal dimension Cannons et al. (2025).

Additionally, the REC models used in our pipeline provide 2D bounding box locations for all expressions in the prompt. This results in redundant information within the cognitive maps, which may confuse or mislead the VLMs. Our ablation studies demonstrated that when the list of objects mentioned in the prompt is known in advance, the RMSE improves by approximately 1.5 meters.

Another limitation lies in the accuracy of metric depth estimation in outdoor environments. To mitigate this, we proposed a relational scaling approach based on common-sense object sizes. However, this method is inherently approximate and not fully reliable. Future work should aim to address these issues by enhancing the depth understanding of VLMs or by improving metric depth estimation in outdoor settings. Better integration of spatial reasoning and scene geometry into VLMs could further improve performance in complex 3D environments. Another major limitation in spatial reasoning datasets and benchmarks is the availability of ground-truth 3D. Previous work has shown the effectiveness of neural radiance fields in generating photo-realistic 3D data Gholami et al. (2024b). Future work should investigate this direction for extending spatial reasoning dataset without having ground-truth 3D annotations.

## A.11 VSI-BENCH'S COGNITIVE MAP VS. EGO3D-VLM:

VSI-Bench uses the VLM itself to estimate a cognitive-map and then passes the generated cognitive-map to the VLM in a second call. However, we use external tools to create the map which leverages higher accuracy in depth estimation and 2D localization of objects. Moreover, creating the cogmap by external tools is faster than calling the VLM twice. In order to quantitatively evaluate, the method in VSI-Bench and Ego3D-VLM, we use VSI-Bench method to create cognitive map and compare its results with Ego3D-VLM (Table 14).

Table 14: Cognitive Map of VSI-Bench vs. Ego3D-VLM

|  | InternVL3-8B | InternVL3-8B$_{\text{VSI-CogMap}}$ | InternVL3-8B$_{\text{Ego3D-VLM}}$ |
| --- | --- | --- | --- |
| Acuracy | 43.1 | 61.6 | 47.0 |

## A.12 FURTHER DETAILS OF THE DATASET

Ego3D-Bench annotations were conducted internally by trained researchers within our organization who are experienced in 3D perception, autonomous driving, and dataset curation. No external crowd workers were used. In total, the annotation process required 112 human-hours. All annotators were compensated according to their standard full-time employment contracts; no task-based or performance-based payment structure was used. Since the annotators were internal researchers rather than external contractors, no additional compensation scheme applied.

Figure 9 shows the distribution of objects in our benchmark. The captioning process was intentionally designed not to rely on rigid linguistic or structural templates. Instead, annotators were instructed to produce concise, factual descriptions that focus on **object identity, 3D spatial attributes**, and **scene-relevant properties** (e.g., "a white sedan parked on the right lane" or "a pedestrian crossing in front of the ego vehicle"). This flexible guideline ensures that captions remain natural while still being compatible with downstream VLM training.

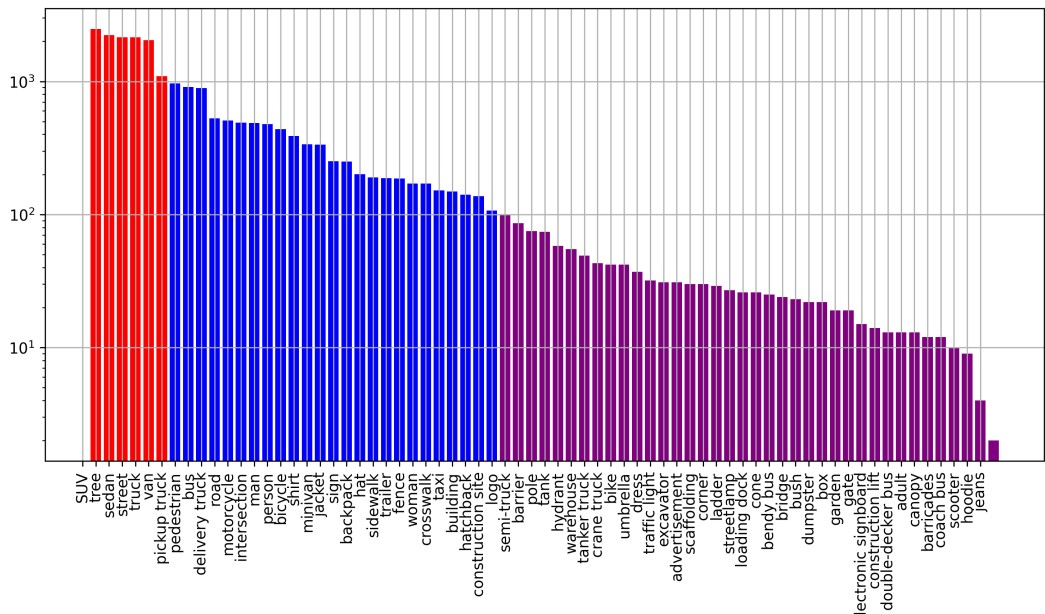

Figure 9: Distribution of objects in Ego3D-Bench.

To maintain **objectivity, reliability, and neutrality**, we incorporated the following quality-control steps:

1. **Clear annotation guidelines:** Annotators were instructed to avoid subjective judgments (e.g., "dangerous driver," "reckless pedestrian"), emotional language, or speculative content. Only directly observable attributes (category, appearance, relative position, size) were allowed.

2. **Structured attribute checklist (soft template):** Although no fixed sentence template was enforced, annotators followed a checklist covering below items which acted as a soft template, improving consistency without forcing unnatural phrasing:

   - Object category
   - Color/shape if visible
   - 3D relation to the ego car (left/right/front/back, distance tier)
   - Action or state (parked, moving, turning, standing still)

### A.13 MULTI-VIEW IMAGE VS. CONCATENATED IMAGES

For generality purposes we used the default setting of passing each view point separately in the main body. In this section we concatenate multi-views into a single image. Table A.12 shows that concatenating multi-views enhances the results of both baseline and our method. We hypothesis that concatenating images reduces the number of vision tokens that enhances models performance.

| | Multiple Images | | Concatenate Images | |
|---|---|---|---|---|
| | InternVL3-8B | InternVL3-8B$_{\text{Ego3D-VLM}}$ | InternVL3-8B | InternVL3-8B$_{\text{Ego3D-VLM}}$ |
| Acuracy | 43.1 | 60.1 | 47.0 | 65.8 |

### A.14 TEXTUAL VS. VISUAL COGNITIVE MAP

Ablation in the main body shows that textual cognitive map outperforms visual cognitive map. We hypothesis that the textual format outperforms the visual format because of the following two reasons. 1) Converting 3D information to image domain and then decoding the information back to textual domain using visual encoders (e.g., siglip) poses errors. However, converting information directly to textual format is lossless. 2) The information density in the textual format is higher than the visual format. In order to quantitatively evaluate this, we have pruned the cogmap tokens and evaluated the performance drop. Table below shows that with the same level of pruning, textual cogmap results in higher performance drop. This result, showcase that the information density in the textual cogmap is higher than visual cogmap. Here the relative accuracy drop is defined as $(\text{Acc}_{\text{ego3d-vlm}} - \text{Acc}_{\text{pruned}})/(\text{Acc}_{\text{ego3d-vlm}} - \text{Acc}_{\text{baseline}})$. It means that the relative accuracy drop in 0% pruning is zero for both methods and in 100% pruning results in 100% relative accuracy drop for both methods.

| CogMap Pruning | Relative Accuracy Drop | |
|---|---|---|
| | Textual-CogMap | Visual-CogMap |
| 0% | 0.0 | 0.0 |
| 10% | 30.2 | 0.0 |
| 20% | 31.0 | 5.3 |
| 30% | 57.5 | 32.3 |

### A.15 ANALYSIS OF ERROR PROPAGATION IN RELATIONAL SCALING

Estimating the relational scaling that was proposed in the main body is prone to errors specifically in the case of partial visibility of objects (e.g. pedestrians), non-upright poses, motion blur, etc. We provide an error propagation analysis showing how variations in scale estimation affect downstream results.

**Theoretical Analysis of Scale Sensitivity to 2D Bounding-Box Errors:** Let $h$ denote the measured bounding-box height in pixels, $z$ the representative object depth, $f_y$ the vertical focal length, and $H$ the assumed real-world object height. Using the pinhole model, the camera-space vertical extent of the object is

$$\Delta y_{3D} = \frac{h \, z}{f_y}.$$

To enforce a metric height of $H$, we estimate the global scale $s$ via

$$s \cdot \Delta y_{3D} = H \quad \Rightarrow \quad \boxed{s = \frac{H f_y}{h \, z}}.$$

The sensitivity of $s$ to small perturbations in the pixel height $h$ follows from

$$\frac{\partial s}{\partial h} = -\frac{H f_y}{h^2 z},$$

so a small error $\delta h$ induces the first-order scale error

$$\delta s = \boxed{-\frac{H f_y}{h^2 z} \delta h}.$$

If we plug the typical values $f_y = 1000$ px, $H = 1.7m$, bbox height $h = 200$ px, and the depth $z = 10m$:

$$\delta s = 0.004 \times \delta h$$

Therefore, the error in scale results the following error in calculating the 3D positions:

$$E = \delta s \times p_{\text{global}} = (0.004 \times \delta h) p_{\text{global}}$$

Therefore, 10% errors in the bounding box (i.e., $\delta h \approx 20$px), results in $E = 0.08 p_{\text{global}}$. It means that for objects let's say 10 meters far from ego the distance error will be $0.08 \times 10 = 0.8m$ and for objects 20 meters far from the ego the error will be 1.6m.

**Empirical Analysis of Scale Sensitivity to 2D Bounding-Box Errors:** We now perform an empirical analysis by adding errors in 2D bounding box detections and calculate the final accuracy of InernVL3-8B$_{\text{Ego3D-VLM}}$ on Ego3D-Bench. At each scale the error was randomly applied to be $+\delta h$ or $+\delta h$ (e.g., $+10\%$ or $-10\%$):

| | Scaling | bbox error | Accuracy |
|---|---|---|---|
| InternVL3-8B | | - | 43.1 |
| InternVL3-8B$_{\text{Ego3D-VLM}}$ | $\times$ | - | 58.4 |
| InternVL3-8B$_{\text{Ego3D-VLM}}$ | $\checkmark$ | $\pm 0\%$ | 60.1 |
| | $\checkmark$ | $\pm 10\%$ | 59.9 |
| | $\checkmark$ | $\pm 20\%$ | 57.6 |
| | $\checkmark$ | $\pm 30\%$ | 55.4 |

## A.16  STATEMENT ON LLMs ASSISTANCE

We declare that some portions of this document have been lightly refined using Large Language Models (e.g., ChatGPT) to enhance clarity and polish. All substantive content and ideas remain entirely our own.

**Ego-Centric Absolute Distance (Meter)**

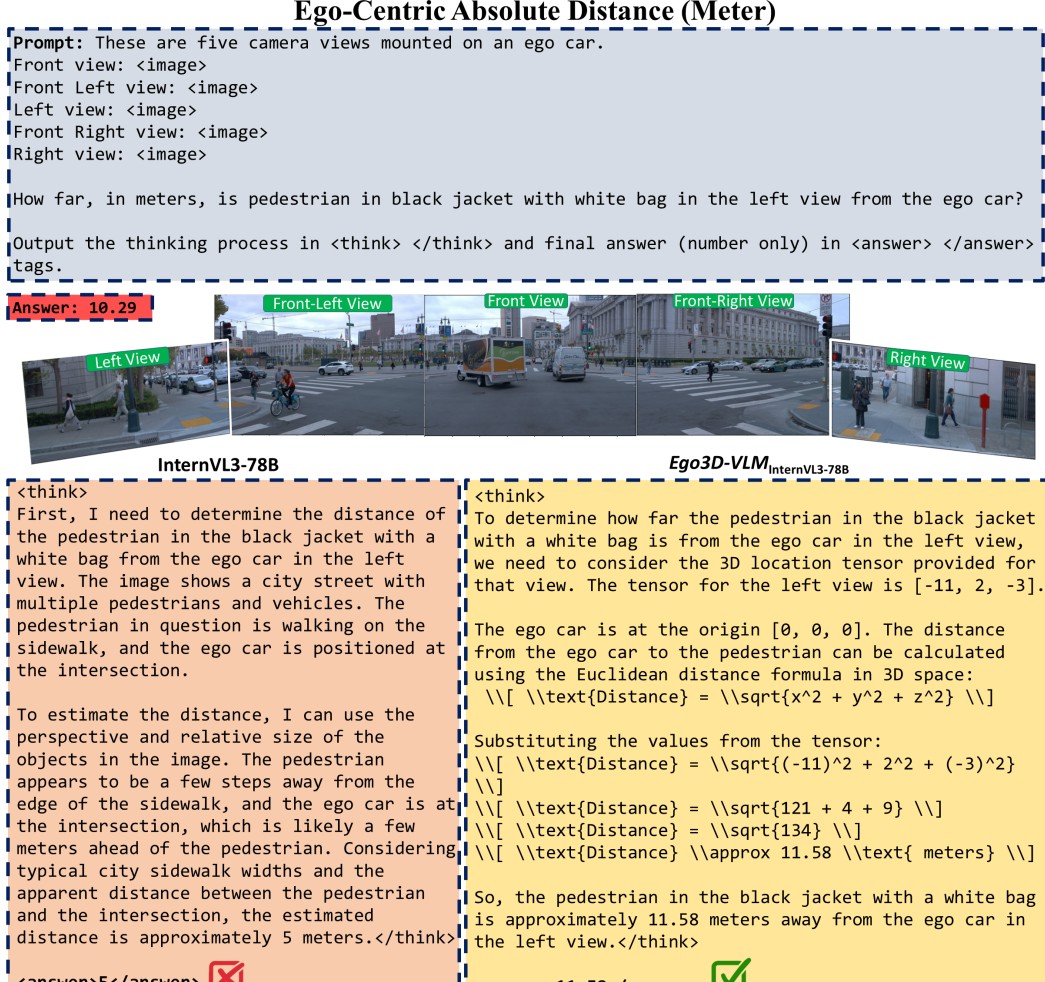

Figure 10: Example responses of the baseline and *Ego3D-VLM* on ego-centric absolute distance task.

## Object-Centric Absolute Distance (Meter)

**Prompt:** These are five camera views mounted on an ego car.
Front view: <image>
Front Left view: <image>
Left view: <image>
Front Right view: <image>
Right view: <image>

How far, in meters, is the postal delivery truck in the front left view from the silver sedan car turning right in the front right view?

Output the thinking process in <think> </think> and final answer (number only) in <answer> </answer> tags.

Answer: 20.05

**InternVL3-78B**

<think>
To determine the distance between the postal delivery truck in the front left view and the silver sedan car turning right in the front right view, we need to consider the layout of the intersection and the positions of the vehicles.

1.**Identify Positions**:
   - The postal delivery truck is on the left side of the intersection, near the curb.
   - The silver sedan is in the middle of the intersection, turning right.
2. **Estimate Intersection Dimensions**:
   - Typical city intersections can vary, but a common width for a city street is about 10-12 meters.
   - Assuming the intersection is roughly square and each lane is about 3 meters wide, the diagonal distance across the intersection can be estimated.
3. **Calculate Distance**:
   - If the truck is near the left curb and the sedan is turning right from the opposite side, they are roughly across the intersection from each other.
   - Using the Pythagorean theorem for a rough estimate, if each side of the intersection is about 10 meters, the diagonal distance would be approximately $\sqrt{10^2 + 10^2} = \sqrt{200} \approx 14.14$ meters.

Given these assumptions, the distance between the two vehicles is approximately 14 meters.\n</think>

<answer>14</answer> ❌

**Ego3D-VLM**_InternVL3-78B_

"<think>
To determine the distance between the postal delivery truck in the front left view and the silver sedan car turning right in the front right view, we need to analyze their respective 3D locations.

1.**Postal Delivery Truck in Front Left View:**
   - 3D location: $([-4, 0, 2])$\n\n2.
     **Silver Sedan Car in Front Right View:**
   - 3D location: $([14, 1, 8])$

We can calculate the Euclidean distance between these two points using the formula:
$$\text{Distance} = \sqrt{(x_2 - x_1)^2 + (y_2 - y_1)^2 + (z_2 - z_1)^2}$$

Plugging in the coordinates:
$$\text{Distance} = \sqrt{(14 - (-4))^2 + (1 - 0)^2 + (8 - 2)^2}$$
$$\text{Distance} = \sqrt{(18)^2 + (1)^2 + (6)^2}$$
$$\text{Distance} = \sqrt{324 + 1 + 36}$$
$$\text{Distance} = \sqrt{361}$$
$$\text{Distance} = 19$$

Therefore, the distance between the postal delivery truck and the silver sedan car is 19 meters.</think>

<answer>19</answer> ✅

Figure 11: Example responses of the baseline and *Ego3D-VLM* on object-centric absolute distance task.

**Ego-Centric Absolute Distance (Multi-Choice)**

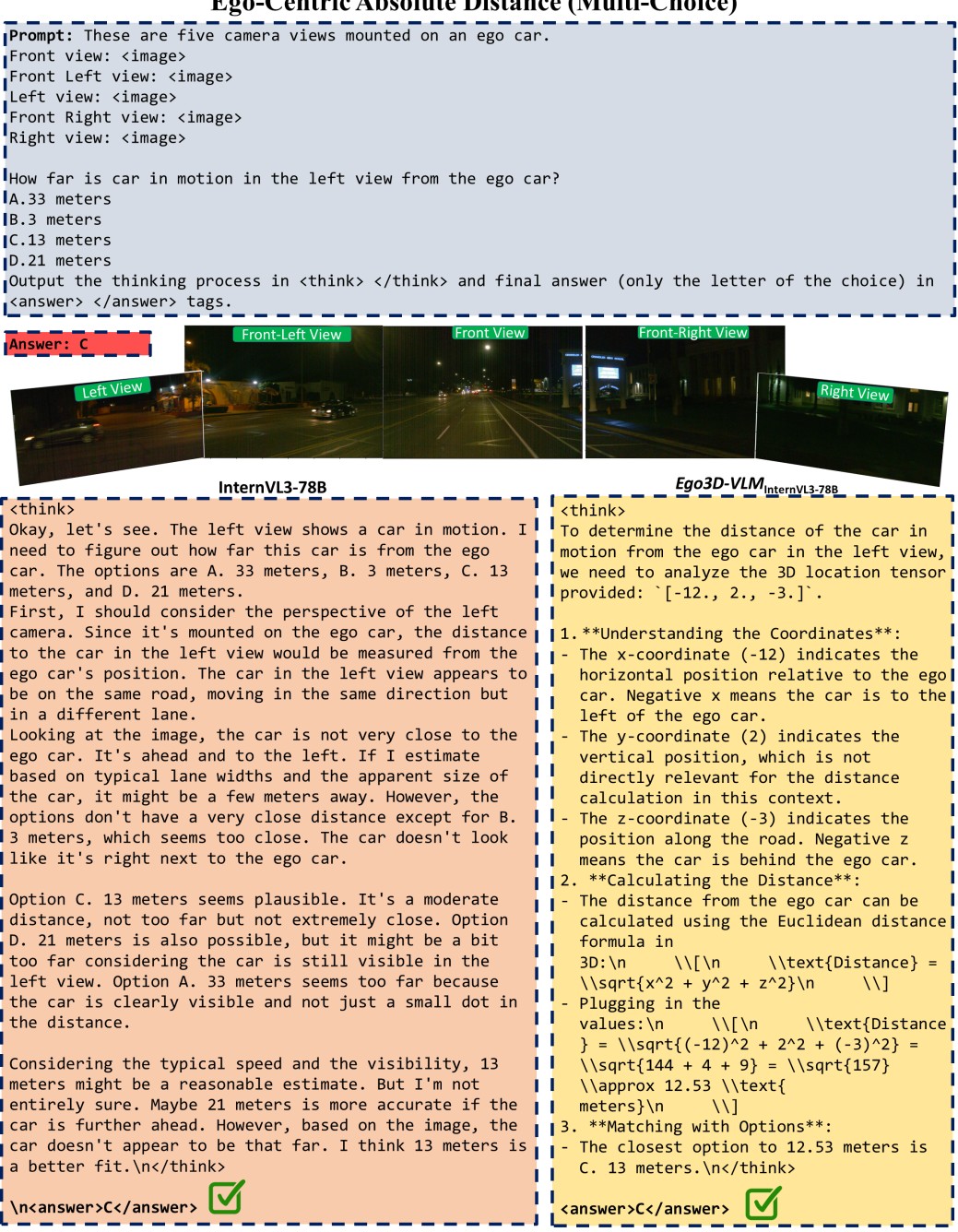

Figure 12: Example responses of the baseline and *Ego3D-VLM* on multi-choice ego-centric absolute distance task.

**Object-Centric Absolute Distance (Multi-Choice)**

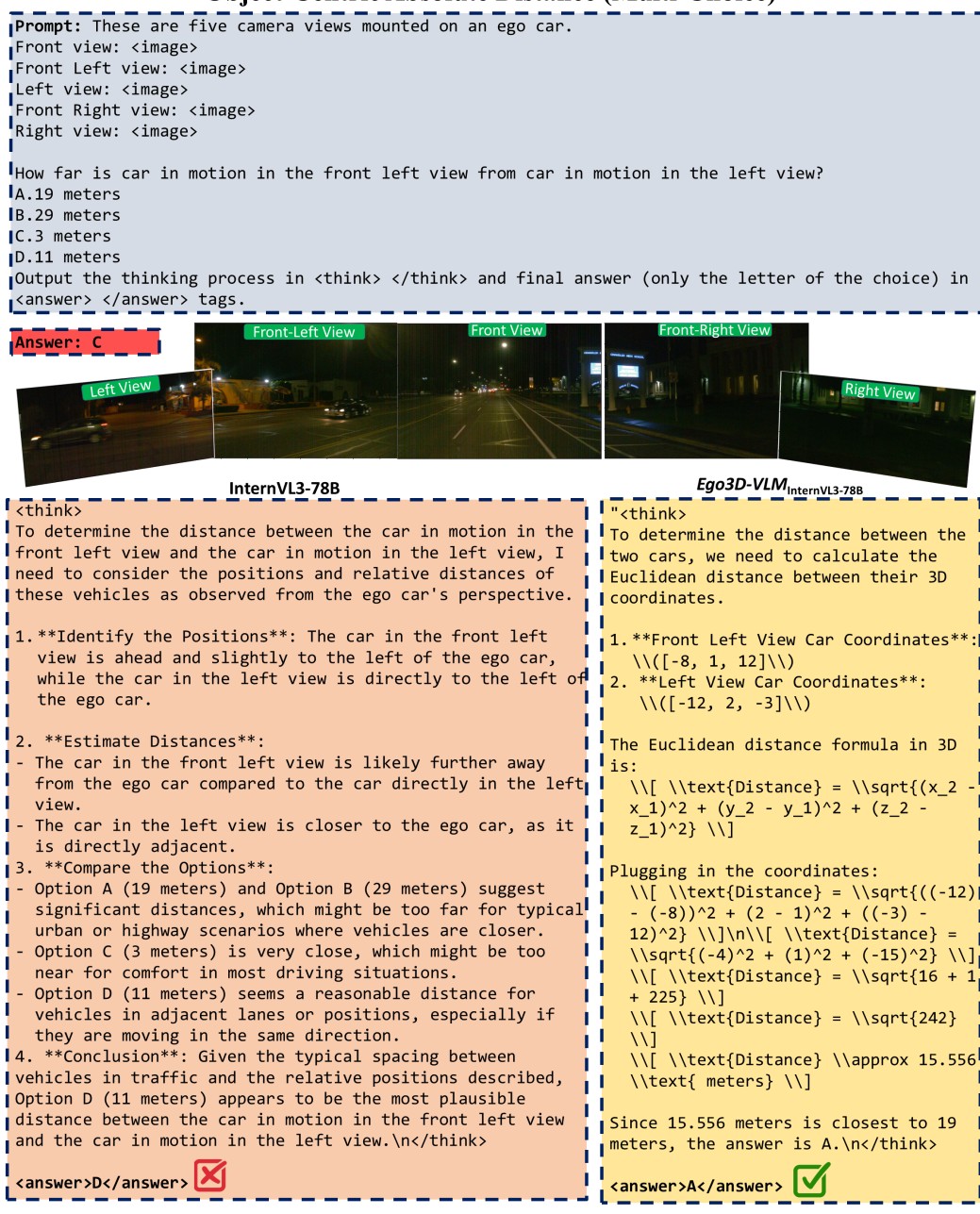

Figure 13: Example responses of the baseline and *Ego3D-VLM* on multi-choice object-centric absolute distance task.

## Object-Centric Motion Reasoning

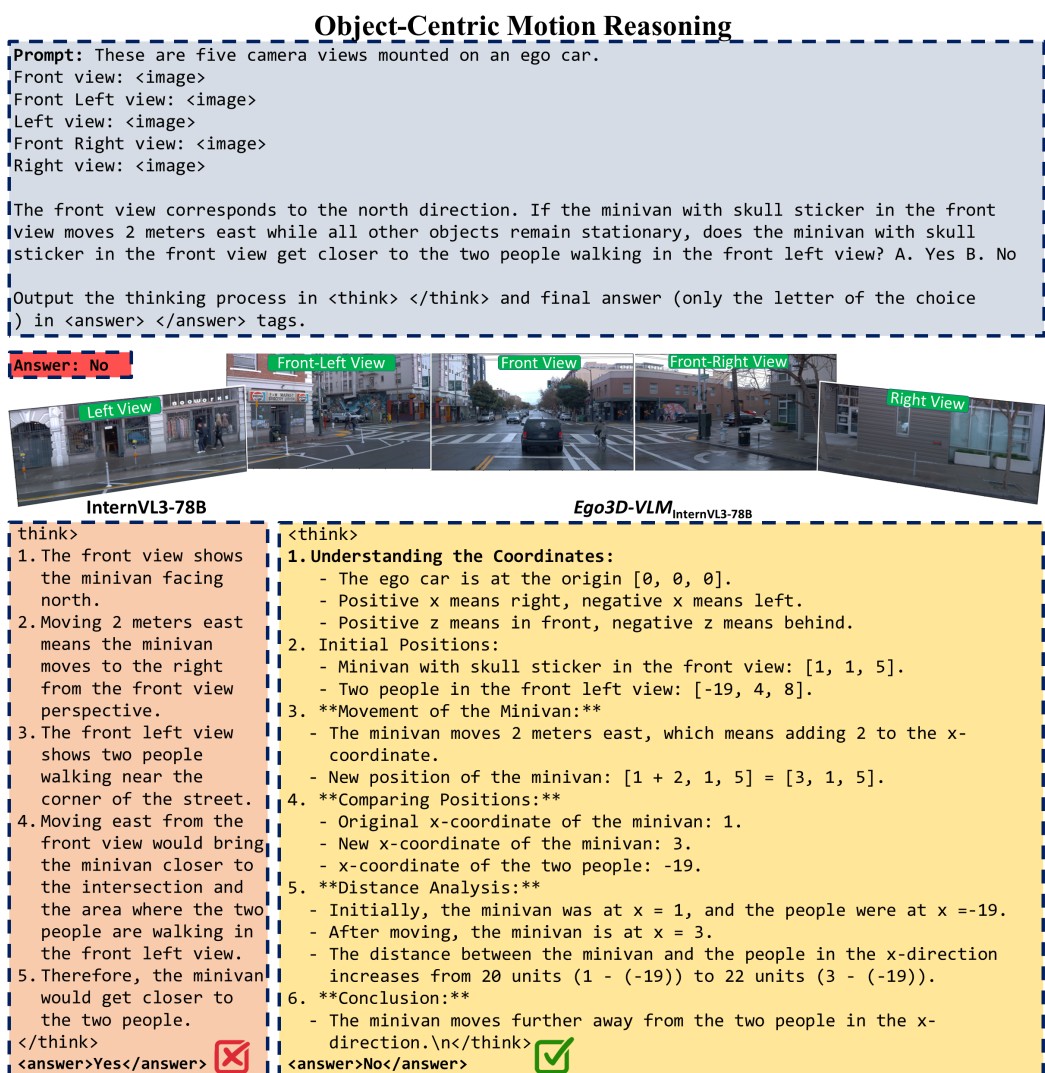

**Prompt:** These are five camera views mounted on an ego car.
Front view: <image>
Front Left view: <image>
Left view: <image>
Front Right view: <image>
Right view: <image>

The front view corresponds to the north direction. If the minivan with skull sticker in the front view moves 2 meters east while all other objects remain stationary, does the minivan with skull sticker in the front view get closer to the two people walking in the front left view? A. Yes B. No

Output the thinking process in <think> </think> and final answer (only the letter of the choice) in <answer> </answer> tags.

Figure 14: Example responses of the baseline and *Ego3D-VLM* on object-centric motion reasoning task.

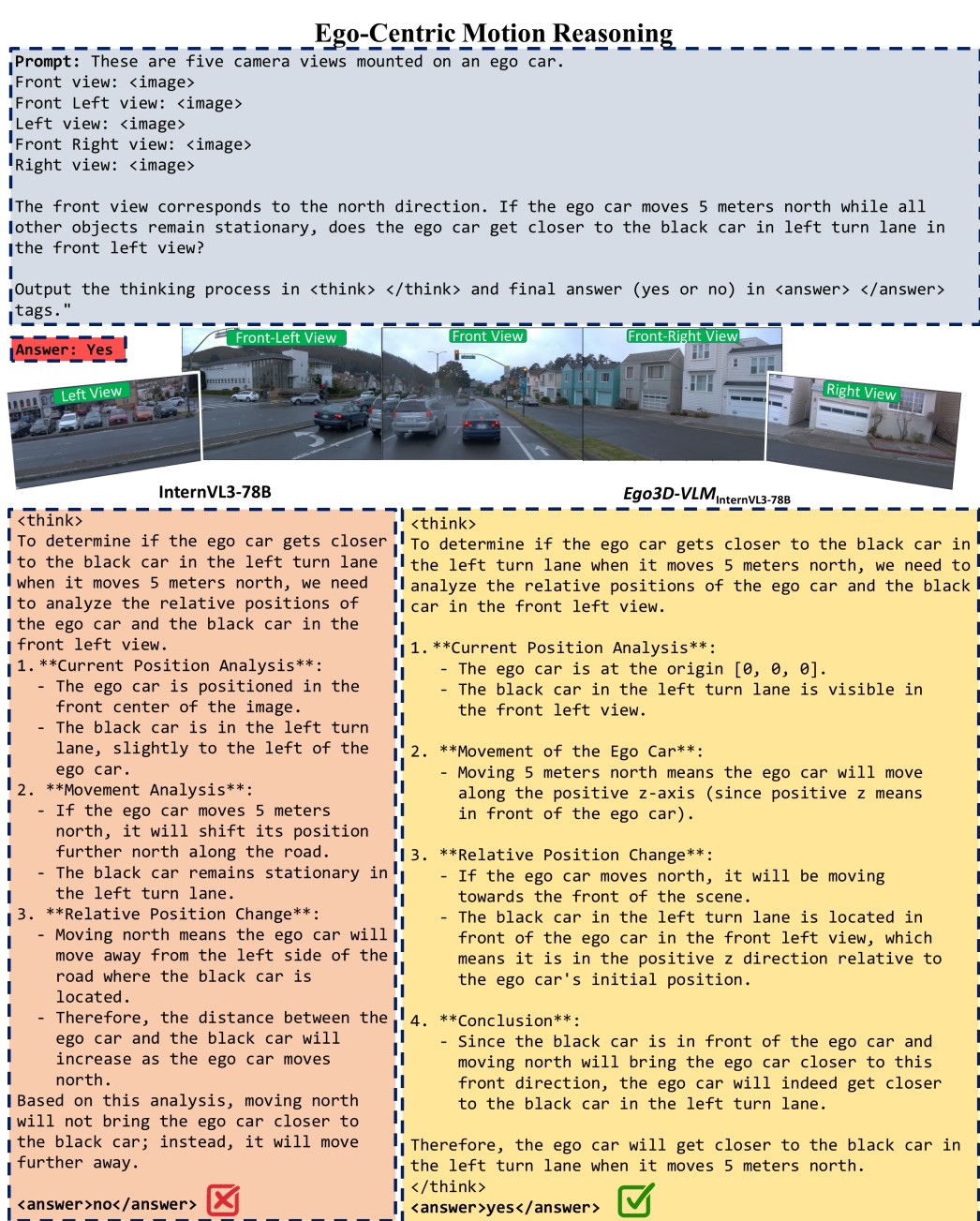

Figure 15: Example responses of the baseline and *Ego3D-VLM* on ego-centric motion reasoning task.

**Travel Time (Ego-Centric)**

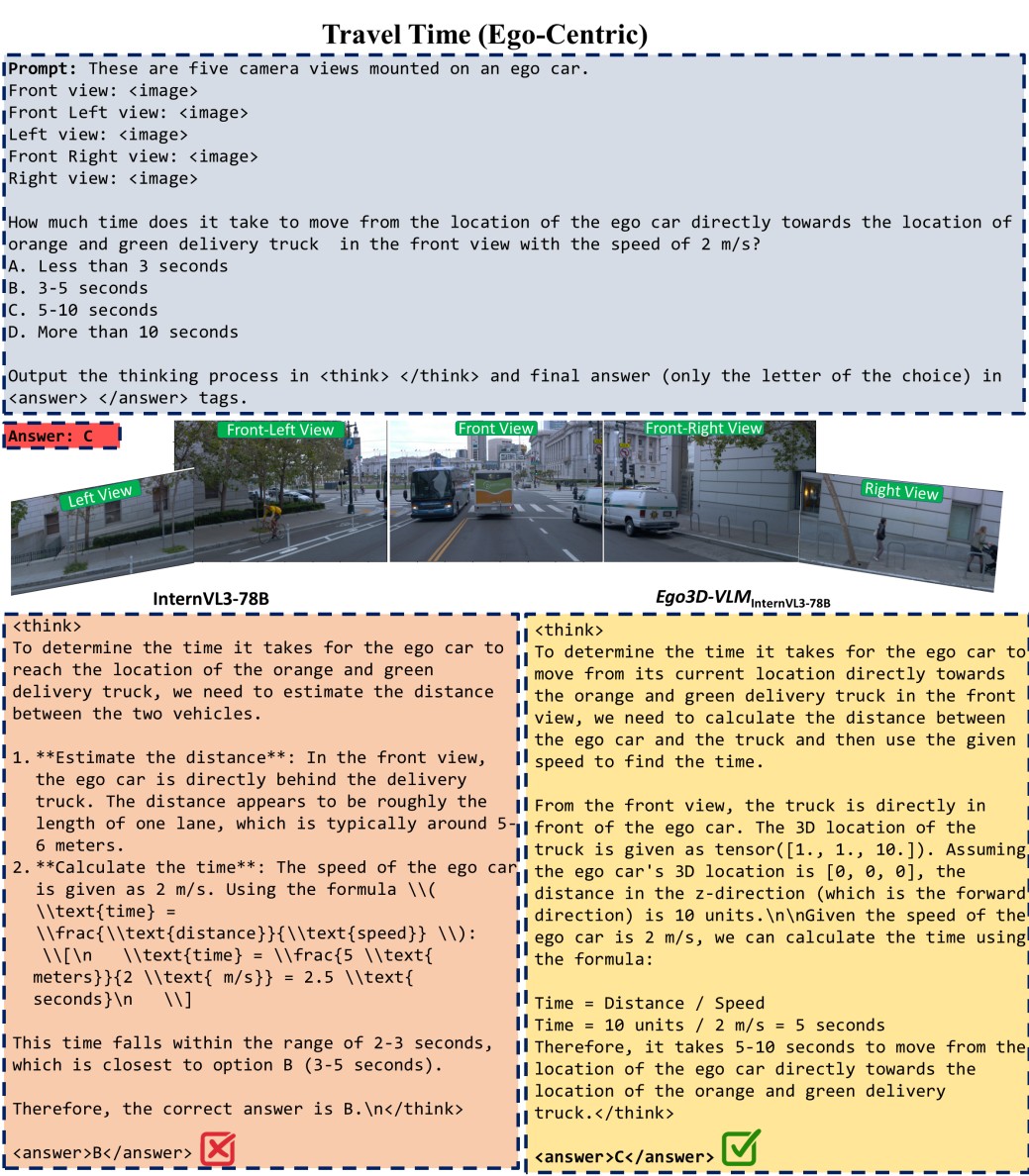

**Prompt:** These are five camera views mounted on an ego car.
Front view: <image>
Front Left view: <image>
Left view: <image>
Front Right view: <image>
Right view: <image>

How much time does it take to move from the location of the ego car directly towards the location of orange and green delivery truck in the front view with the speed of 2 m/s?
A. Less than 3 seconds
B. 3-5 seconds
C. 5-10 seconds
D. More than 10 seconds

Output the thinking process in <think> </think> and final answer (only the letter of the choice) in <answer> </answer> tags.

**Answer: C**

**InternVL3-78B**

<think>
To determine the time it takes for the ego car to reach the location of the orange and green delivery truck, we need to estimate the distance between the two vehicles.

1. **Estimate the distance**: In the front view, the ego car is directly behind the delivery truck. The distance appears to be roughly the length of one lane, which is typically around 5-6 meters.
2. **Calculate the time**: The speed of the ego car is given as 2 m/s. Using the formula \\( \\text{time} = \\frac{\\text{distance}}{\\text{speed}} \\):
 \\[\n   \\text{time} = \\frac{5 \\text{ meters}}{2 \\text{ m/s}} = 2.5 \\text{ seconds}\n   \\]

This time falls within the range of 2-3 seconds, which is closest to option B (3-5 seconds).

Therefore, the correct answer is B.\n</think>

<answer>B</answer> ❌

**Ego3D-VLM**~InternVL3-78B~

<think>
To determine the time it takes for the ego car to move from its current location directly towards the orange and green delivery truck in the front view, we need to calculate the distance between the ego car and the truck and then use the given speed to find the time.

From the front view, the truck is directly in front of the ego car. The 3D location of the truck is given as tensor([1., 1., 10.]). Assuming the ego car's 3D location is [0, 0, 0], the distance in the z-direction (which is the forward direction) is 10 units.\n\nGiven the speed of the ego car is 2 m/s, we can calculate the time using the formula:

Time = Distance / Speed
Time = 10 units / 2 m/s = 5 seconds
Therefore, it takes 5-10 seconds to move from the location of the ego car directly towards the location of the orange and green delivery truck.</think>

<answer>C</answer> ✅

Figure 16: Example responses of the baseline and *Ego3D-VLM* on travel time (ego-centric) task.

## Travel Time (Object-Centric)

**Prompt:** These are five camera views mounted on an ego car.
Front view: <image>
Front Left view: <image>
Left view: <image>
Front Right view: <image>
Right view: <image>

How much time does it take to move from the location of the white delivery van in the front view directly towards to the location of the pedestrian standing in brown jacket in the right view with the speed of 2 m/s?
A. Less than 3 seconds
B. 3-5 seconds
C. 5-10 seconds
D. More than 10 seconds

Output the thinking process in <think> </think> and final answer (only the letter of the choice) in <answer> </answer> tags.

**Answer: C**

**InternVL3-78B**

<think>
To determine the time it takes to move from the location of the white delivery van in the front view to the location of the pedestrian in the right view, we need to estimate the distance between these two points and then use the given speed of 2 m/s.

1. **Estimate the distance:**
   - The front view shows the van at an intersection.
   - The right view shows the pedestrian on the sidewalk near a building.
   - Visually, the distance appears to be roughly the length of a city block, which is typically around 100 meters.
2. **Calculate the time:**
   - Using the formula: time = distance/speed
   - Distance = 100 meters
   - Speed = 2 m/s
   - Time = 100 meters / 2 m/s = 50 seconds
   - Given the options, 50 seconds falls into the \"More than 10 seconds\" category.
Therefore, the correct answer is D.\n</think>

<answer>D</answer> ❌

**Ego3D-VLMInternVL3-78B**

<think>
To determine the time it takes for the ego car to move from the location of the white delivery van in the front view to the location of the pedestrian in the right view, we need to calculate the distance between these two points and then use the given speed of 2 m/s.

1. **Identify the 3D locations:**
   - White delivery van in the front view: [15, 2, 20]
   - Pedestrian in the right view: [8, 2, 3]
2. **Calculate the distance between the two points:**
   - The x-coordinate difference: |15 - 8| = 7 meters
   - The z-coordinate difference: |20 - 3| = 17 meters
   - The y-coordinate difference is 0, so it does not affect the distance.
3. **Use the Pythagorean theorem to find the distance:**
   - Distance = $\sqrt{7^2 + 17^2}$ = $\sqrt{49 + 289}$ = $\sqrt{338}$ $\approx$ 18.38 meters
4. **Calculate the time:**
   - Time = Distance / Speed = 18.38 meters / 2 m/s ~= 9.19 seconds

Since 9.19 seconds falls within the range of 5-10 seconds, the correct answer is C.\n</think>

<answer>C</answer> ✅

Figure 17: Example responses of the baseline and *Ego3D-VLM* on travel time (object-centric) task.

## Object-Centric Relative Distance

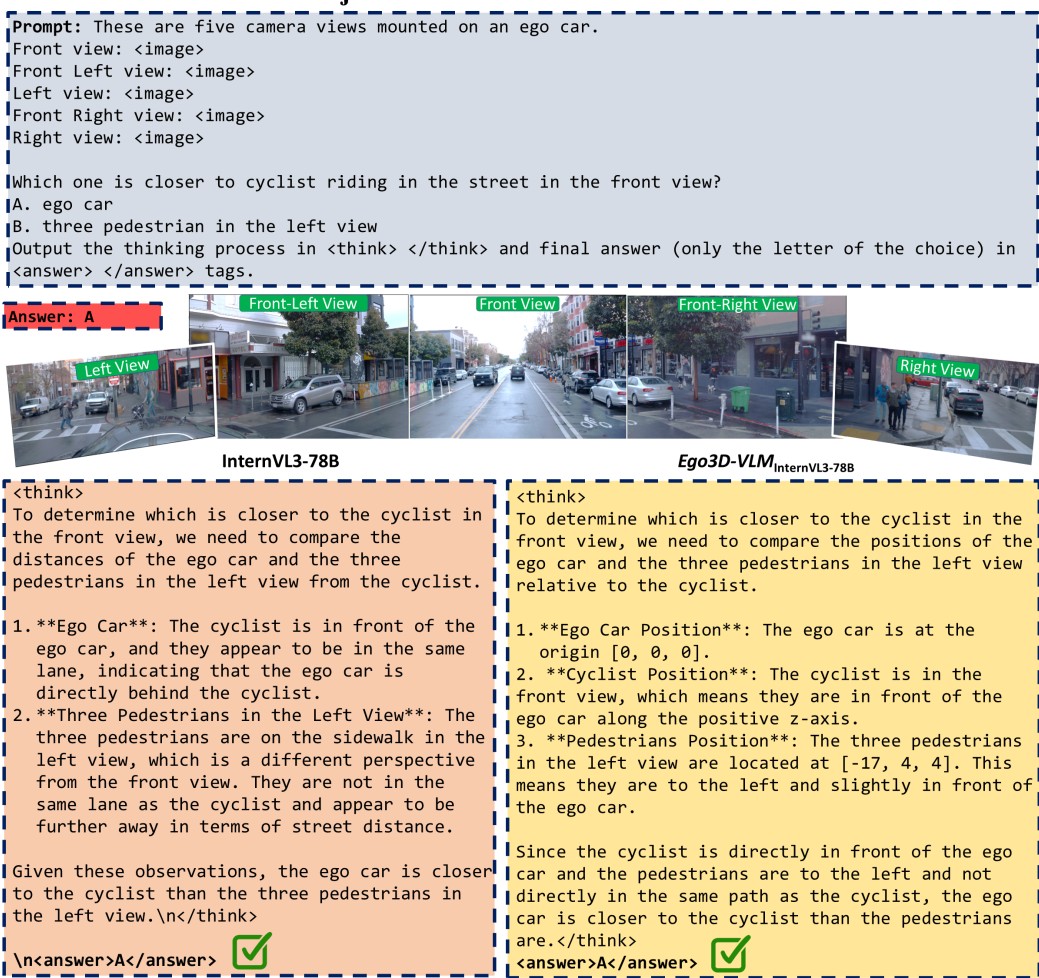

**Prompt:** These are five camera views mounted on an ego car.
Front view: <image>
Front Left view: <image>
Left view: <image>
Front Right view: <image>
Right view: <image>

Which one is closer to cyclist riding in the street in the front view?
A. ego car
B. three pedestrian in the left view
Output the thinking process in <think> </think> and final answer (only the letter of the choice) in
<answer> </answer> tags.

**Answer: A**

**InternVL3-78B**

<think>
To determine which is closer to the cyclist in the front view, we need to compare the distances of the ego car and the three pedestrians in the left view from the cyclist.

1.**Ego Car**: The cyclist is in front of the ego car, and they appear to be in the same lane, indicating that the ego car is directly behind the cyclist.
2.**Three Pedestrians in the Left View**: The three pedestrians are on the sidewalk in the left view, which is a different perspective from the front view. They are not in the same lane as the cyclist and appear to be further away in terms of street distance.

Given these observations, the ego car is closer to the cyclist than the three pedestrians in the left view.\n</think>

\n<answer>A</answer> ☑

**Ego3D-VLM**InternVL3-78B

<think>
To determine which is closer to the cyclist in the front view, we need to compare the positions of the ego car and the three pedestrians in the left view relative to the cyclist.

1.**Ego Car Position**: The ego car is at the origin [0, 0, 0].
2. **Cyclist Position**: The cyclist is in the front view, which means they are in front of the ego car along the positive z-axis.
3. **Pedestrians Position**: The three pedestrians in the left view are located at [-17, 4, 4]. This means they are to the left and slightly in front of the ego car.

Since the cyclist is directly in front of the ego car and the pedestrians are to the left and not directly in the same path as the cyclist, the ego car is closer to the cyclist than the pedestrians are.</think>
<answer>A</answer> ☑

Figure 18: Example responses of the baseline and *Ego3D-VLM* on object-centric relative distance task.

**Ego-Centric Relative Distance**

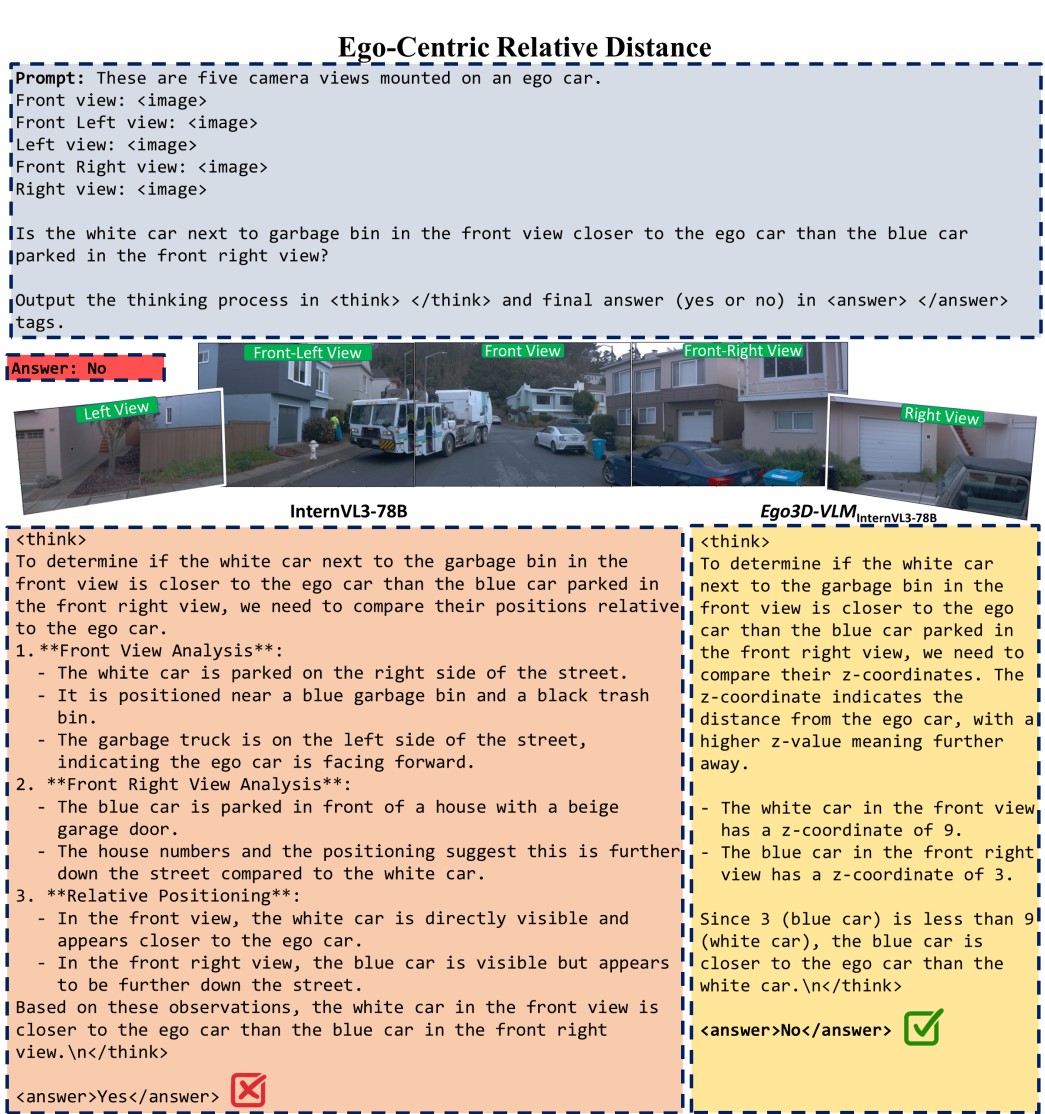

Figure 19: Example responses of the baseline and *Ego3D-VLM* on ego-centric relative distance task.

## Localization

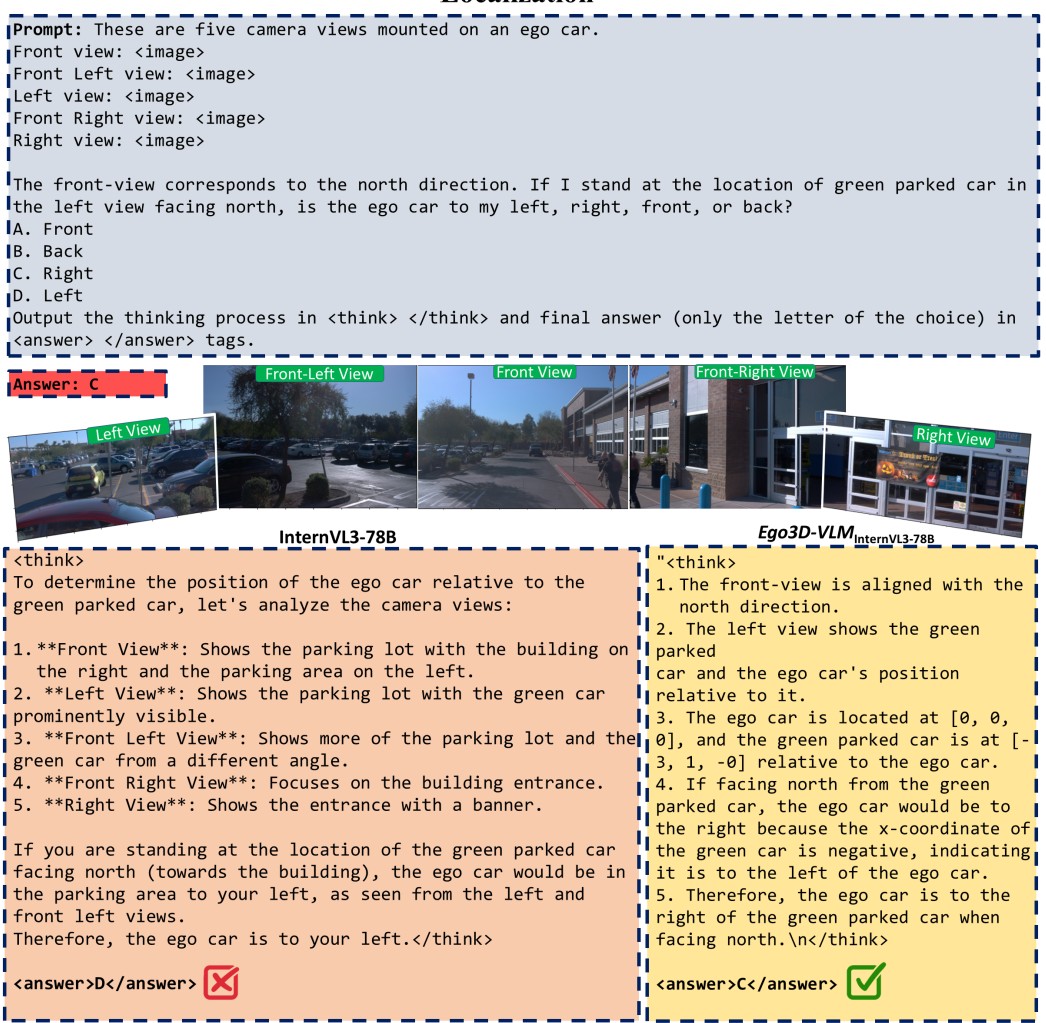

**Prompt:** These are five camera views mounted on an ego car.
Front view: `<image>`
Front Left view: `<image>`
Left view: `<image>`
Front Right view: `<image>`
Right view: `<image>`

The front-view corresponds to the north direction. If I stand at the location of green parked car in the left view facing north, is the ego car to my left, right, front, or back?
A. Front
B. Back
C. Right
D. Left
Output the thinking process in `<think> </think>` and final answer (only the letter of the choice) in `<answer> </answer>` tags.

Answer: C

**InternVL3-78B**

```
<think>
To determine the position of the ego car relative to the
green parked car, let's analyze the camera views:

1.**Front View**: Shows the parking lot with the building on
   the right and the parking area on the left.
2. **Left View**: Shows the parking lot with the green car
prominently visible.
3. **Front Left View**: Shows more of the parking lot and the
green car from a different angle.
4. **Front Right View**: Focuses on the building entrance.
5. **Right View**: Shows the entrance with a banner.

If you are standing at the location of the green parked car
facing north (towards the building), the ego car would be in
the parking area to your left, as seen from the left and
front left views.
Therefore, the ego car is to your left.</think>

<answer>D</answer> ❌
```

**Ego3D-VLM_InternVL3-78B**

```
"<think>
1.The front-view is aligned with the
   north direction.
2. The left view shows the green
parked
car and the ego car's position
relative to it.
3. The ego car is located at [0, 0,
0], and the green parked car is at [-
3, 1, -0] relative to the ego car.
4. If facing north from the green
parked car, the ego car would be to
the right because the x-coordinate of
the green car is negative, indicating
it is to the left of the ego car.
5. Therefore, the ego car is to the
right of the green parked car when
facing north.\n</think>

<answer>C</answer> ✅
```

Figure 20: Example responses of the baseline and *Ego3D-VLM* on localization task.

