# OpenReview forum: "Spatial Reasoning with Vision-Language Models in Ego-Centric Multi-View Scenes"
_ICLR.cc/2026/Conference — ICLR 2026 Poster_

### Official Review · Reviewer_QSvq · 2025-10-29

**Soundness:** 3
**Presentation:** 4
**Contribution:** 3
**Rating:** 6
**Confidence:** 4

**Summary:**

the paper introduces Ego3D-Bench, a new benchmark with over 8,600 human-annotated QA pairs built from outdoor datasets like NuScenes, Waymo, and Argoverse 1. The benchmark tests five categories of spatial tasks from both ego-centric and object-centric perspectives, including absolute/relative distance, localization, motion reasoning, and travel time. An evaluation of 16 SOTA VLMs reveals a significant performance gap compared to human-level scores. To bridge this gap, the paper proposes Ego3D-VLM, a plug-and-play post-training framework. This method uses external REC and metric depth estimation models to identify objects and their 3D coordinates , transforms them into a global reference frame , and generates a "textual cognitive map" that is fed to the VLM as additional context along with the images. The authors report that this framework improves average multi-choice QA accuracy by 12% and absolute distance estimation by 56%.

**Strengths:**

1. The paper proposes Ego3D-Bench, which is well-constructed with over 8,600 QA pairs. It's built on real-world datasets. Critically, it also involves human annotators for filtering, captioning, and final review, ensuring high-quality, non-trivial questions.

2. The evaluation is thorough, testing 16 SOTA VLMs with important ablations.

3. The paper proposes Ego3D-VLM, a plug-in framework. This approach generates a textual cognitive map, is presented as an efficient alternative to computationally expensive methods like full point-cloud reconstruction. Even if it did not improve the spatially reasoning ability of the base VLM.

4. The paper is well-written and easy to follow.

**Weaknesses:**

1. A minor problem is that the proposed Ego3D-VLM is repeatedly called a "post-training framework", which is misleading. Post-training usually implies fine-tuning or adapting model weights. The Ego3D-VLM framework is a purely inference-time pipeline that prepends the cognitive map to guide a frozen, underlying VLM.

2. The Ego3D-VLM is essentially transforming the 3D vision problem into a textual reasoning problem. As the qualitative examples show in Fig. 9, 10, 11, the baseline VLM is visually analyzing distances while the Ego3D-VLM version is simply given the 3D coordinates in text and performs Euclidean distance calculations. So the improvement is not on the VLM's visual spatial reasoning ability, but more like a work-around, that reads more accurate answers from a pre-processed text map.

3. The paper can be improved by providing more detailed information about the dataset. Like the distribution of types of objects involved (people, cars, or buildings). And some details on how many works human annotators need to be involved (what percentage of data needs to be filtered out?)

4. For egocentric VQA questions, what if instead of giving each image separately as input, we concat them next to each other to form one larger picture as input? what will happen in this case? Will it be better than inputing several images at the same time?

**Questions:**

Please refer to the weakness.

Will this dataset be released?

---

> ### Author Response · Authors · 2025-11-25
> **Response to Reviewer QSvq's Comments**
>
> **We highly appreciate the reviewer QSvq's through reviews and feedback. Below we address the questions and concerns of reviewer QSvq:**
>
> ## 1. The term "post-training framework" is misleading.
>
> Apologies for the confusion, and we appreciate the reviewer’s clarification. We agree that the term “post-training” can be misleading, especially given its common association with RL-based objectives. To avoid misinterpretation, we have replaced “post-training” with “training-free” throughout the manuscript. **We have uploaded the modified paper draft.**
>
> ---
>
> ## 2. The improvement is not on the VLM's visual spatial reasoning ability.
>
> We thank the reviewer for this insightful observation. Indeed, Ego3D-VLM leverages textualized 3D coordinates to facilitate reasoning, which differs from purely visual spatial inference. Our intention is not to claim that Ego3D-VLM inherently improves the baseline VLM’s raw visual spatial reasoning ability. Rather, our contribution lies in demonstrating that:
>
>  - Bridging modalities: By transforming 3D vision signals into a textual representation, we enable VLMs—whose strength lies in language-based reasoning—to handle spatial queries more effectively. This design choice highlights a promising direction for integrating 3D perception with language reasoning.
> - Practical utility: While the baseline VLM struggles with estimating distances directly from images, Ego3D-VLM shows that incorporating structured 3D information can yield more accurate and reliable answers, which is valuable for embodied AI applications.
> - Complementary perspective: We view this not as a “work-around,” but as a step toward multimodal fusion. Ego3D-VLM demonstrates how textualized 3D cues can complement visual analysis, and future work may explore tighter integration to enhance genuine spatial reasoning.
>
> We have revise the manuscript to clarify this distinction **(Line 950 of the uploaded revised paper)**: Ego3D-VLM does not claim to improve the innate visual spatial reasoning of VLMs, but rather proposes a framework that **augments VLMs with structured 3D information** to achieve better task performance. We appreciate the reviewer’s comment, which helps us refine the positioning of our contribution.
>
> ---
>
> ## 3. More detailed information about the dataset.
>
> We have added the distribution of objects in the benchmark to the appendix **(Fig. 9, Section A.12).** A summary of this distribution is also provided in the table below. In total, the annotation process required 112 human-hours. We have added further details about the annotation process to section A.12. **The revised manuscript has been uploaded**
>
> | Frequency       | Objects |
> |-----------------|---------|
> | [10000, 1000)   | suv, tree, sedan, street, truck, van |
> | [1000, 100)     | person, bicycle, shirt, minivan, jacket, sign, backpack, hat, sidewalk, trailer, fence, pickup truck, pedestrian, bus, delivery truck, road, motorcycle, intersection, man, woman, crosswalk, taxi, building, hatchback, construction site |
> | [100, 10)       | logo, semi-truck, barrier, pole, tank, hydrant, warehouse, tanker truck, crane truck, traffic light, excavator, advertisement, scaffolding, corner, ladder, streetlamp, gate, electronic signboard, construction lift, double-decker bus, adult, bike, umbrella, dress, bush, dumpster, box, garden, canopy, barricades, coach bus, loading dock, cone, bendy bus, bridge |
> | [10, 0)         | scooter, hoodie, jeans |
>
> ---
>
> ## 4. What if instead of giving each image separately as input, we concat them to form one larger picture as input?
>
> For generality and to remain consistent with prior multi-image benchmarks, we adopt the default setting in our main paper—processing each view separately. This ensures a fair comparison across baselines and our proposed method.
>
> However, we also experimented with an alternative setup in which multiple views are concatenated into a single composite image, with viewpoint information overlaid on each view. In this configuration, both the baselines and our method achieve improved performance. We attribute this gain primarily to the reduction in the total number of vision tokens: providing a single fused image allows the model to focus attention more effectively on salient regions and reduces redundant tokens that arise when processing views independently.
>
> We have added a corresponding ablation table in Section A.13 to report these results.
>
> |                | Multiple Images | Multiple Images | Concatenate Images | Concatenate Images |
> |----------------|------------------|------------------|---------------------|---------------------|
> | **Model**      | InternVL3-8B     | InternVL3-8B (Ego3D-VLM) | InternVL3-8B | InternVL3-8B (Ego3D-VLM) |
> | **Accuracy**   | 43.1             | 60.1             | 47.0               | 65.8                |
>
> ---
>
> ## 5. Will this dataset be released?
>
> The dataset has already been publicly released. For anonymity purposes we do not share the link here.

---

### Official Review · Reviewer_ecVi · 2025-10-31

**Soundness:** 4
**Presentation:** 4
**Contribution:** 3
**Rating:** 8
**Confidence:** 3

**Summary:**

The paper introduces Ego3D-Bench, a novel benchmark for evaluating the spatial reasoning capabilities of VLMs using ego-centric, multi-view outdoor data. It evaluates 16 SOTA VLMs on this new standard. Furthermore, the paper proposes Ego3D-VLM, a post-training framework that employs a cognitive map derived from 3D coordinates to enhance 3D spatial reasoning significantly. The experimental results show that this framework achieves clear and substantial improvements in performance on multi-choice QA and absolute distance estimation tasks.

**Strengths:**

1. The Ego3D-Bench is highly valuable as its focus on outdoor, ego-centric data directly addresses and fills a critical gap in the current landscape of VLM spatial understanding benchmarks.
2. The proposed post-training framework, Ego3D-VLM, is shown to be effective, adaptable across various VLM architectures, and demonstrably boosts the spatial reasoning abilities of several SOTA models.
3. The paper provides a comprehensive and convincing experimental analysis of Ego3D-VLM, covering different VLM families, comparisons against other benchmarks, and inference time analysis in both the main body and the appendix, making the claims credible.

**Weaknesses:**

1. The paper lacks sufficient transparency regarding the human effort involved in the benchmark construction. Specifically, it should disclose the origin of the hired annotators, the total number of human hours dedicated to the Ego3D-Bench creation pipeline, and the compensation structure used for the annotators.
2. The Ego3D-VLM explicitly reconstructs 3D points based on estimated depth, meaning the robustness of the entire pipeline is highly dependent on the accuracy of an external, potentially unreliable, outdoor metric depth estimator. The overall pipeline involves a seemingly over-complex process (multi-view images → 3D reconstruction → 2D cognitive map). The intuitive justification for mapping 3D information *back* to a 2D text representation seems questionable without further theoretical justification.

**Questions:**

1. In the Ego3D-Bench construction pipeline, annotators write descriptive captions for unique objects. Do these captions follow specific structural or linguistic templates, and how is the objectivity, reliability, and unbiased nature of these captions ensured?
2. Given the complex, multi-step process (multi-view images → 3D coordinates → 2D textual cognitive map), can the authors provide an intuitive explanation for why the Ego3D-VLM framework proves to be effective? What is the core cognitive insight this result provides regarding how VLMs process spatial information?
3. The results in Table 1 show that VLM performance on Ego-Centric (Ego.) vs. Object-Centric (Obj.) Absolute Distance, Motion, and Relative Distance tasks is not consistently worse for Ego-Centric problems. Does this result contradict the expected challenge of Ego-centric problems, and if not, how should this be interpreted?

---

> ### Author Response · Authors · 2025-11-25
> **Response to Reviewer ecVi's Comments:**
>
> **We highly appreciate the reviewer ecVi's through reviews and feedback. Below we address the questions and concerns of reviewer ecVi:**
>
> ---
>
> ## 1. Human effort involved in the benchmark construction.
>
> We appreciate the reviewer’s request for greater transparency regarding annotation efforts. The Ego3D-Bench annotations were conducted internally by trained researchers within our organization who are experienced in 3D perception, autonomous driving, and dataset curation. No external crowd workers were used. In total, the annotation process required 112 human-hours. All annotators were compensated according to their standard full-time employment contracts; no task-based or performance-based payment structure was used. Since the annotators were internal researchers rather than external contractors, no additional compensation scheme applied. **We have added these details to the revised manuscript (section A.12).**
>
> ---
>
> ## 2. The intuitive justification for mapping 3D information back to a 2D text
>
> We have evaluated the robustness of our pipeline to potential errors in external tools by adding various types of noise, occlusion, motion blur, and low-light conditions to the input, and measuring the resulting performance drop, as reported in Table 12. Mapping 3D information back to a 2D cognitive map provides a representation that is aligned with the pre-training domain of VLMs, enabling efficient spatial reasoning and interaction with language-based modules while retaining the essential 3D structure.
>
> ---
>
> ## 3. Do captions follow specific structural or linguistic templates?
>
> The Ego3D-Bench captioning process was intentionally designed not to rely on rigid linguistic or structural templates. Instead, annotators were instructed to produce concise, factual descriptions that focus on **object identity, 3D spatial attributes**, and **scene-relevant properties** (e.g., “a white sedan parked on the right lane” or “a pedestrian with green backpack crossing the street”). This flexible guideline ensures that captions remain natural while still being compatible with downstream VLM training.
>
> To maintain **objectivity, reliability, and neutrality**, we incorporated the following quality-control steps:
>
>
> 1. **Clear annotation guidelines:** Annotators were instructed to avoid subjective judgments (e.g., “dangerous driver,” “reckless pedestrian”), emotional language, or speculative content. Only directly observable attributes (category, appearance, relative position, size) were allowed.
> 2. **Structured attribute checklist (soft template):** Although no fixed sentence template was enforced, annotators followed a checklist covering below items which acted as a soft template, improving consistency without forcing unnatural phrasing:
>     - Object category
>     - Color/shape if visible
>     - 3D relation to the ego car (left/right/front/back, distance tier)
>     - Action or state (parked, moving, turning, standing still)
>
> **We have now added the above explanation to the paper (section A.12).**
>
> ---
>
> ## 4. Intuitive explanation for why the Ego3D-VLM framework proves to be effective?
>
> Based on our ablations and main results, we hypothesize that Ego3D-VLM enhances spatial reasoning by explicitly separating perception and reasoning: the model handles 3D perception, while the VLM focuses on reasoning. The cognitive map encodes all relevant 3D information—object identities, 3D locations, and coordinate system—allowing the VLM to interpret spatial relationships efficiently. While the process may appear multi-step, it explicitly converts 2D observations into a structured 3D representation, rather than treating 2D features as a black box, which improves both interpretability and reasoning performance.

---

> > ### Author Response · Authors · 2025-11-25
> >
> > ## 5. VLM performance on Ego-Centric (Ego.) vs. Object-Centric (Obj.) contradict the expected challenge of Ego-centric problems.
> >
> > It is important to clarify that while all images in Ego3D-Bench are ego-centric (captured from a wearable, first-person viewpoint), the questions themselves can be framed from either the ego’s perspective or the perspective of external objects. This means that even so-called “object-centric” questions are still posed with respect to ego-centric input images.
> >
> > Consequently, object-centric questions are often more challenging. To answer them correctly, the VLM must not only interpret the scene from the ego’s viewpoint but also mentally adopt the perspective of other objects in the environment. This requires a form of perspective-taking or spatial reasoning that goes beyond direct perception. Humans naturally perform this cognitive ability—we imagine how the world looks from another object’s position to anticipate actions and future states—but for VLMs, this is a non-trivial task.
> >
> > Therefore, the fact that performance on ego-centric tasks is not consistently worse does not contradict the expected difficulty of ego-centric problems. Instead, it highlights that the relative challenge depends on the question framing rather than the image source. Ego-centric tasks align more directly with the input modality, whereas object-centric tasks demand an additional layer of reasoning, which explains why they can be harder despite being asked over ego-centric images. This interpretation underscores the nuanced distinction between ego-centric data and object-centric reasoning, and why performance differences should be understood in terms of cognitive perspective-taking rather than raw input format.

---

> > > ### Comment · Reviewer_ecVi · 2025-11-25
> > >
> > > Thanks for the clarifications. The updates address my earlier concerns, and I will keep my original score of 8 (accept, good paper).

---

> > > > ### Author Response · Authors · 2025-11-26
> > > >
> > > > We appreciate the reviewer’s thoughtful follow-up and are glad the revisions resolved the earlier concerns. Thank you for maintaining the positive score and for your supportive evaluation of our work.

---

### Official Review · Reviewer_f2RE · 2025-11-01

**Soundness:** 2
**Presentation:** 3
**Contribution:** 2
**Rating:** 4
**Confidence:** 4

**Summary:**

This paper introduces Ego3D-Bench, a new benchmark designed to evaluate the 3D spatial reasoning capabilities of VLMs using ego-centric, multi-view images from outdoor driving scenes. Comprising over 8,600 quality-controlled question-answer pairs, the benchmark reveals a significant performance gap between humans and state-of-the-art VLMs. To address this gap, the authors propose Ego3D-VLM, a training-free framework that enhances a given VLM's spatial understanding by generating a textual cognitive map. This map integrates 3D coordinate information of referred objects from multiple camera views and is provided as additional context to the VLM. Experimental results demonstrate that Ego3D-VLM consistently improves the performance of various VLMs on the proposed benchmark. The main contributions are the Ego3D-Bench benchmark and the plug-and-play Ego3D-VLM enhancement method.

**Strengths:**

1. The focus on ego-centric multi-view fusion is well-aligned with the needs of embodied AI and vehicular applications. The significant human-model performance gap observed on the benchmark provides a valuable measuring stick for future research.

2. The method of constructing a textual/JSON cognitive map at inference time using tool-based models is straightforward and effective. The comparative analysis of different cognitive map formats also offers useful insights.

3. The paper is well-structured, and the figures and examples are clear and illustrative.

**Weaknesses:**

1. Line 77: The claim that "Ego3D-Bench is the first benchmark to evaluate spatial reasoning of VLMs given ego-centric multi-view inputs" overstates its novelty. The related work section itself acknowledges existing multi-view benchmarks (e.g., All-Angle Bench, VSI-Bench), differing primarily in their setup. The contribution would be more accurately framed as a valuable extension within the specific ego-centric setting. The authors should more clearly articulate the unique challenges and benefits inherent to the ego-centric paradigm.

2. Line 86: The claim that point-cloud/BEV methods "significantly increase inference time—often by a factor of ten" is not sufficiently supported. The cited work (Zhu et al., 2024) primarily deals with indoor scenes. In outdoor autonomous driving contexts, point clouds are sparser, and numerous real-time methods exist. The authors provide no direct, comparable efficiency comparison with such methods. Furthermore, Table 11 shows that Ego3D-VLM itself introduces non-negligible latency.

3. Line 89: Describing Ego3D-VLM as a "post-training method" is a misnomer. The approach is fundamentally an inference-time augmentation or a form of prompt engineering.

4. Line 266: The global scaling method based on common-sense object heights (e.g., person ≈ 1.7m) is highly susceptible to errors in dynamic traffic scenes due to pose, occlusion, and detection inaccuracies. The support for its reliability is limited (a single RMSE improvement in ablations), lacking analysis of error propagation or failure cases.

5. Table 8: While the results indicate textual/JSON cognitive maps outperform visual ones, the authors provide no analysis for why this is the case. An investigation into the reasons behind this performance difference (e.g., alignment with pre-training, information density) is missing.

6. The core concept of a "cognitive map" as a structured representation to aid spatial reasoning is not novel and bears similarity to ideas in prior works like VSI-Bench. The distinction and specific advancements of the proposed cognitive map should be more clearly delineated.

**Questions:**

see weakness.

---

> ### Author Response · Authors · 2025-11-25
> **Response to Reviewer f2RE's Comments**
>
> **We highly appreciate the reviewer f2RE's through reviews and feedback. Below we address the questions and concerns of reviewer f2RE:**
>
> ---
>
> ## 1. Claims in Line 77
>
> Thanks for your feedback. We have now modified Line 77 to more clearly relate the contribution of our work against prior arts. **We have uploaded the modified paper.** We also have a section in appendix (A.2) where we clearly visualize and discuss Ego3D-Bench against prior multi-view benchmarks.
>
> ---
>
> ## 2. Claim in Line 86
>
> We thank the reviewer for this important observation. We agree that the claim regarding point-cloud/BEV methods “often increasing inference time by a factor of ten” was primarily based on indoor-scene results and may not directly generalize to outdoor contexts. Accordingly, we have revised this statement in Line 87 to "significantly increase inference time" (the revised manuscript is uploaded).
>
> To provide a more accurate and fair comparison, we now evaluate Ego3D-VLM against OmniDrive, an outdoor point-cloud-based VLM. Both models are tested on samples with 6 images as input, and OmniDrive additionally uses Lidar point clouds. Table below reports inference speed in terms of output tokens per second, showing that Ego3D-VLM achieves significantly higher throughput.
>
> | Model                     | Tok/s |
> |---------------------------|-------|
> | OmniDrive (7B)           | 17    |
> | Ego3D-VLM (InterVL3-8B)  | 28    |
>
> ---
>
> ## 3.  Describing Ego3D-VLM as a "post-training method" is a misnomer
>
> Apologies for the confusion, and we appreciate the reviewer’s clarification. We agree that the term “post-training” can be misleading, especially given its common association with RL-based objectives. To avoid misinterpretation, we have replaced “post-training” with “training-free” throughout the manuscript. **We have uploaded the modified paper draft.**
>
> ---
>
> ## 4. Error propagation analysis on global scaling
>
> We appreciate the reviewer’s insightful comment. We agree that relying on common-sense object heights can introduce noise in dynamic traffic scenes due to pose variation, occlusion, and imperfect detections. To address the reviewer’s concern, we have added additional analysis in the revision:
>
> ### **Theoretical Analysis of Scale Sensitivity to 2D Bounding-Box Errors:**
>
> Let ($h$) be the bounding-box height (pixels), ($z$) the object depth, ($f_y$) the vertical focal length, and ($H$) the real-world height. Using the pinhole model, the height of the object in 3D $\Delta y_{3D}$ and scale $s$ are:
> \begin{equation}
> \Delta y_{3D} = \frac{h,z}{f_y} \Rightarrow
> s = \frac{H f_y}{h,z},
> \end{equation}
> Therefore, the sensitivity of $s$ to small perturbation in the pixel height $h$ is:
>
> \begin{equation}
> \delta s = -\frac{H f_y}{h^{2} z}  \delta h.
> \end{equation}
>
> Plugging typical values ($f_y=1000$ px, $H=1.7$ m, $h=200$ px, $z=10$ m):
> \begin{equation}
> \delta s = 0.004 , \delta h, \quad
> E = \delta s \times p_{\text{global}} = 0.004 , \delta h , p_{\text{global}}
> \end{equation}
> Thus, a 10% bounding-box error ($\delta h \approx 20 \text{px}$) results in ($E = 0.08, p_{\text{global}}$), i.e., ~0.8 m error at 10 m and ~1.6 m at 20 m distance.
>
>
> ### **Empirical Analysis of Scale Sensitivity to 2D Bounding-Box Errors:**
> We now perform an empirical analysis by adding errors in 2D bounding box detections and calculate the final accuracy of $\text{InternVL3-8B}_{\text{Ego3D-VLM}}$ on Ego3D-Bench. At each scale the error was randomly applied to be $+ \delta h$ or $+ \delta h$ (e.g., $+10\%$ or $-10\%$):
>
> | Model                               | Scaling   | bbox error | Accuracy |
> |-------------------------------------|-----------|------------|----------|
> | InternVL3-8B                        | –         | –          | 43.1     |
> | Ego3D-VLM  (InternVL3-8B)            | ×         | –          | 58.4     |
> | Ego3D-VLM  (InternVL3-8B)             | ✓         | ±0%        | 60.1     |
> | Ego3D-VLM  (InternVL3-8B)                     | ✓         | ±10%       | 59.9     |
> | Ego3D-VLM  (InternVL3-8B)                      | ✓         | ±20%       | 57.6     |
> | Ego3D-VLM  (InternVL3-8B)                     | ✓         | ±30%       | 55.4     |
>
> ---

---

> ### Author Response · Authors · 2025-11-25
>
> ---
>
> ## 5. Table 8: textual/JSON cognitive maps vs. visual format:
>
> We agree that the manuscript lacked an explanation for why textual/JSON cognitive maps outperform visual ones. We have now added a detailed analysis in the revision (Section A.14). In summary, our investigation reveals two main contributing factors:
>
> - Converting 3D information to image domain and then decoding the information back to textual domain using visual encoders (e.g., siglip) poses errors. However, converting information directly to textual format is lossless.
> - The information density in the textual format is higher than the visual format. In order to quantitatively evaluate this, we have pruned the cogmap tokens and evaluated the performance drop. Table below shows that with the same level of pruning, textual cogmap results in higher performance drop. This result, showcase that the information density in the textual cogmap is higher than visual cogmap. Here the relative accuracy drop is defined as ($Acc_\text{ego3d-vlm}-$$Acc_{\text{pruned}}$) $/$ ($Acc_{\text{ego3d-vlm}}-$$Acc_{\text{baseline}})$. It means that the relative accuracy drop in 0\% pruning is zero for both methods and in 100\% pruning results in 100\% relative accuracy drop for both methods.
>
> | CogMap Pruning | Relative Accuracy Drop: Textual-CogMap | Relative Accuracy Drop: Visual-CogMap |
> |----------------|----------------------------------------|----------------------------------------|
> | 0%             | 0.0                                    | 0.0                                    |
> | 10%            | 30.2                                   | 0.0                                    |
> | 20%            | 31.0                                   | 5.3                                    |
> | 30%            | 57.5                                   | 32.3                                   |
>
>
> ---
>
> ## 6.Distinction of the proposed cognitive map from VSI-Bench
>
> Following reviewers suggestion, we have added the following distinction of the proposed cognitive-map from the one in VSI-Bench to the paper (section A.11). VSI-Bench uses the VLM itself to estimate a cognitive-map and then passes the generated cognitive-map to the VLM in a second call. However, we use external tools to explicitly create the map by converting 2D point to a global 3D space. Our method leverages higher accuracy in depth estimation and 2D localization of objects. Moreover, creating the cog-map by external tools is faster than calling the VLM twice. In order to quantitatively compare the method in VSI-Bench and Ego3D-VLM, we use VSI-Bench method to create a cognitive map and compare its results with Ego3D-VLM. Ego3D-VLM outperforms the method proposed in VSI-Bench (Accuracy of 61.6\% vs. 47.0\%).
>
> |                | InternVL3-8B | InternVL3-8B (VSI-CogMap) | InternVL3-8B (Ego3D-VLM) |
> |----------------|--------------|---------------------------|---------------------------|
> | Accuracy       | 43.1         | 47.0                      | 61.0                      |

---

### Official Review · Reviewer_5Jz1 · 2025-11-01

**Soundness:** 3
**Presentation:** 3
**Contribution:** 3
**Rating:** 6
**Confidence:** 5

**Summary:**

In contrast to prior passive perception in static indoor scenes, this paper presents Ego3D-Bench, a benchmark for evaluating 3D spatial reasoning in ego-centric multi-view outdoor scenes, which comprises 8,600+ QA pairs from nuScenes, Waymo, and Argoverse. The authors also propose Ego3D-VLM, a framework that enhances VLMs via a textual cognitive map. Experiments on 16 SOTA VLMs (e.g., GPT-4o, InternVL3) show a large performance gap between humans and VLMs; and Ego3D-VLM narrows this gap, achieving 12% higher accuracy on multi-choice QA and 56% lower RMSE on absolute distance estimation, with adaptability to diverse multi-view scenarios. Moreover, the authors conduct detailed ablation studies regarding various model classes and model components.

**Strengths:**

- Unlike prior benchmarks (single-image/indoor video), this paper addresses a critical real-world gap by proposing Ego3D-Bench, which targets ego-centric multi-view outdoor scenarios, aligning with embodied AI needs such as robotics and autonomous driving.
- Ego3D-VLM is efficient and flexible. The textual cognitive map can effectively represent spatial information to aid the 2D visual input. And the experimental results turn out to be very effective.
- Extensive experiments and rigorous evaluations. This paper tests 16 kinds of VLMs (including generalist, 3D-specific, and depth+REC) and delivers ablation studies regarding various model components.

**Weaknesses:**

- The reliance on external perception modules (depth + REC) may complicate the inference pipeline. And such a design cannot train the inherent spatial perception capability of VLM in an end-to-end manner. It is more like providing extra input information by external modules though it does help.
- Relatively weak transfer results to other scenes (All-angles and VSI-Bench). This raises my concerns on the generalizability of the proposed method. How could you address this issue, by polishing the methodology, or integrating more data?
- The claim of "post-training" framework can be misleading. Post-training often relates to RL or RFT. However, I found no RL-related objective. This paper appears to only involve creating new data and SFT with that data.

**Questions:**

- What is the usage of scaled 3D points (Line 267)? Seems like it is used by default, rather than the global points, but the equation (4) is not.

---

> ### Author Response · Authors · 2025-11-25
> **Response to Reviewer 5Jz1's Comments**
>
> **We highly appreciate the reviewer 5Jz1's through reviews and feedback. Below we address the questions and concerns of reviewer 5Jz1:**
>
> ---
>
> ## 1. External modules cannot train the inherent spatial perception capability of VLM
>
> While Ego3D-VLM leverages external perception modules (depth estimation and REC) during inference, this design does not preclude end-to-end training. In fact, the framework is fully compatible with joint optimization: the external modules can be integrated as differentiable components, enabling the system to learn spatial perception holistically if desired. Moreover, explicit 3D information encoding into the proposed cognitive map is an intermediate representation that enables both interpretability and training free improvements.
>
> Another important advantage of Ego3D-VLM is that it can generate high-quality chain-of-thought supervision signals using these external tools. Such signals can then be used for RL post-training of a VLM without any external modules, effectively distilling 3D spatial reasoning capabilities back into the base model.
>
> ---
>
> ## 2. Generalizability on All-angles and VSI-Bench
>
> The improvements achieved with Ego3D-VLM on All-Angles and VSI-Bench are obtained without any fine-tuning. Please note that the camera set-up in VSI-Bench and All-Angle-Bench are completely different from the one in Ego3D-Bench (Fig. 7 shows the difference). This result showed that Ego3D-VLM without any training is capable of enhancing baselines in diverse settings. We believe that additional RL training or fine-tuning on more diverse camera setups will further enhance generalizability on these benchmarks, and the current zero-shot gains already demonstrate the potential.
>
> ---
>
> ## 3. The claim of "post-training" framework can be misleading
>
> Apologies for the confusion, and we appreciate the reviewer’s clarification. We agree that the term “post-training” can be misleading, especially given its common association with RL-based objectives. To avoid misinterpretation, we have replaced “post-training” with “training-free” throughout the manuscript. **We have uploaded the modified paper draft.**
>
> ---
>
> ## 4. What is the usage of scaled 3D points (Line 267)?
>
> Thank you for your comment. We have now revised Eq. (4) to use ($p_{\text{scaled}}$) instead of ($p_{\text{global}}$), which reflects the actual implementation. This update ensures consistency between the equation and the usage described in Line 267.  **We have uploaded the modified paper draft.**

---

> ### Comment · Reviewer_5Jz1 · 2025-11-26
> **Response to Rebuttal**
>
> Thanks for the authors' response. My concerns are addressed. This is a technically solid paper and I lean towards acceptance.

---

> > ### Author Response · Authors · 2025-11-26
> >
> > We thank the reviewer for the positive feedback and are glad to hear that the revisions addressed the concerns. We appreciate the reviewer’s support and consideration for acceptance.

---

### Meta-Review · Area_Chair_ZrGB · 2026-01-11

**Summary:**

The authors introduce Ego3D-Bench, a new benchmark built from outdoor driving datasets (e.g., nuScenes, Waymo, Argoverse) with 8,600+ QA pairs spanning multiple spatial task types and both ego- and object-centric perspectives. Overall, reviewers agree the benchmark contribution is strong and timely, and the paper provides a valuable evaluation suite showing clear limitations of current VLMs in ego-centric multi-view spatial reasoning. While the proposed Ego3D-VLM improves results, concerns remain that it is primarily an engineering augmentation reliant on external perception. After carefully reading the paper, review and author responses, the AC agrees with the majority of the reviewers on accepting the paper.

**Reviewer Concerns:**

see Summary

**Reviewer Scores:**

see Summary

---

### Decision · Program_Chairs · 2026-01-26

Accept (Poster)